# MiR-21 Regulates Growth and Migration of Cervical Cancer Cells by RECK Signaling Pathway

**DOI:** 10.3390/ijms25074086

**Published:** 2024-04-06

**Authors:** Seidy Y. Aguilar-Martínez, Gabriela E. Campos-Viguri, Selma E. Medina-García, Ricardo J. García-Flores, Jessica Deas, Claudia Gómez-Cerón, Abraham Pedroza-Torres, Elizabeth Bautista-Rodríguez, Gloria Fernández-Tilapa, Mauricio Rodríguez-Dorantes, Carlos Pérez-Plasencia, Oscar Peralta-Zaragoza

**Affiliations:** 1Direction of Chronic Infections and Cancer, Research Center in Infection Diseases, Instituto Nacional de Salud Pública, Cuernavaca 62100, Mexico; yudith.aguilar.m@gmail.com (S.Y.A.-M.); gaby_gecv@hotmail.com (G.E.C.-V.); sel.medgar@gmail.com (S.E.M.-G.); jafet.floresgar@gmail.com (R.J.G.-F.); jessicadeas@gmail.com (J.D.); 2Department of Epidemiology of Cancer, Research Center Population Health, Instituto Nacional de Salud Pública, Cuernavaca 62100, Mexico; cceron@insp.mx; 3Programa Investigadoras e Investigadores por México, Consejo Nacional de Humanidades, Ciencias y Tecnologías, México City 14080, Mexico; abraneet@gmail.com; 4Hereditary Cancer Clinic, Instituto Nacional de Cancerología, México City 14080, Mexico; 5Clinical Chemistry, Faculty of Health Sciences, Universidad Autónoma de Tlaxcala, Zacatelco 90750, Mexico; eli.bautista@gmail.com; 6Clinical Research Laboratory, Faculty of Chemical Biological Sciences, Universidad Autónoma de Guerrero, Chilpancingo 39070, Mexico; gferti@hotmail.com; 7Oncogenomics Laboratory, Instituto Nacional de Medicina Genómica, Tlalpan, México City 14610, Mexico; chynos@yahoo.com; 8Oncogenomics Laboratory, Instituto Nacional de Cancerología, México City 14080, Mexico; carlos.pplas@gmail.com; 9Biomedicine Unit, FES-Iztacala, Universidad Nacional Autónoma de México, Tlalnepantla de Baz 54090, Mexico

**Keywords:** cervical cancer, HPV, microRNAs, miR-21, RECK, siRNAs

## Abstract

Expression of miR-21 has been found to be altered in almost all types of cancers, and it has been classified as an oncogenic microRNA. In addition, the expression of tumor suppressor gene RECK is associated with miR-21 overexpression in high-grade cervical lesions. In the present study, we analyze the role of miR-21 in RECK gene regulation in cervical cancer cells. To identify the downstream cellular target genes of upstream miR-21, we silenced endogenous miR-21 expression using siRNAs. We analyzed the expression of miR-21 and RECK, as well as functional effects on cell proliferation and migration. We found that in cervical cancer cells, there was an inverse correlation between miR-21 expression and RECK mRNA and protein expression. SiRNAs to miR-21 increased luciferase reporter activity in construct plasmids containing the RECK-3′-UTR microRNA response elements MRE21-1, MRE21-2, and MRE21-3. The role of miR-21 in cell proliferation was also analyzed, and cancer cells transfected with siRNAs exhibited a markedly reduced cell proliferation and migration. Our findings indicate that miR-21 post-transcriptionally down-regulates the expression of RECK to promote cell proliferation and cell migration inhibition in cervical cancer cell survival. Therefore, miR-21 and RECK may be potential therapeutic targets in gene therapy for cervical cancer.

## 1. Introduction

Despite advancements in screening and vaccination, cervical cancer continues to pose a significant public health challenge, ranking as a major cause of oncological morbidity and mortality among women globally. This disparity is particularly pronounced in low- and middle-income countries, where access to preventive and therapeutic interventions remains limited [1,2]. Infection by high-risk oncogenic HPV is associated with the development of precancerous lesions and anogenital carcinomas, currently at the forefront of prophylactic and therapeutic strategy development. Additionally, HPV contributes to head and neck cancers, broadening the spectrum of malignancies linked to this virus and highlighting the necessity for comprehensive management and treatment approaches [3]. Interestingly, the National Health Service England first aimed to tackle the burden of the disease by introducing a national cervical screening program in 1988, which has since seen a significant reduction in over a third of cases in England [4]. Cervical carcinoma has the capacity to metastasize to distant tissues and organs, a fact strongly associated with treatment failure. However, different research groups are conducting ongoing immunotherapy trials to evaluate vaccine-based therapies, adoptive T-cell therapy, and immune-modulating agents in patients with advanced cervical cancer [5]. Transformation of non-invasive cervical neoplasia into an invasive cervical carcinoma requires the migration of epithelial cells through the subjacent extracellular matrix, a process in which the matrix metalloproteinases (MMPs) play a critical role by degrading the extracellular matrix components [6,7]. MMPs are also dependent on local equilibrium with their physiologic inhibitors, such as the RECK protein (reversion-inducing cysteine-rich protein with kazal motifs), which primarily regulates MMPs’ activities. Interestingly, an imbalance of the MMPs/RECK axis has been associated with high-risk oncogenic HPV infection in high-grade cervical lesions [6].

Initially, the RECK gene was identified in v-Ki-Ras-transformed NIH 3T3 cells during the induction of a flat morphology, and it was thought to be a tumor suppressor gene that suppressed invasion [8]. Currently, we now know that the RECK gene is located on the 9p13-p12 chromosomal locus, has 21 exons with 87 kb of length, and is considered as a tumor suppressor gene [9]. The protein encoded by this gene is a glycosylphosphatidylinositol (GPI) membrane-anchored glycoprotein protease regulator of 971 amino acids of 110 kDa that contains serine-proteinase-inhibitor-like domains. RECK is widely expressed in normal human tissues and is downregulated in a wide variety of tumor tissues, including cervical cancer, and its under-expression often correlates with poorer prognoses [10,11,12]. Induced expression of RECK results in suppression of tumor angiogenesis, invasion, and metastasis [13]. This evidence supports a role for RECK in the regulation in vivo of MMPs which are known to be involved in cancer progression and implicate RECK downregulation in tumor angiogenesis. RECK mutations, however, are rare in cancer genomes, suggesting that agents that re-activate dormant RECK may be of clinical value [14].

Although RECK has different signaling pathways of transcriptional regulation [15], accumulating evidence supports the role of microRNAs in RECK regulation during the carcinogenesis processes. It has been reported that RECK is regulated by miR-15a in neuroblastoma [16], miR-15b/16, miR-21, miR-372, and miR-373 in colon adenocarcinoma [17], miR-25 and miR-374b-5p in gastric cancer [18,19], miR-92a, miR-92b, and miR-96 in lung cancer [20,21,22], miR-135b and sponging miR-135b in hepatocellular carcinoma [23,24], miR-182-5p in prostate cancer, bladder cancer, breast cancer, and melanoma [25,26,27,28], miR-181a-5p, miR-200b, miR-200c, and miR-221 in colorectal cancer [29,30,31], miR-15b-5p in prostate cancer [32], miR-9 and miR-590-5p in oral squamous cell carcinoma [33,34], miR-544a in endometrial carcinoma [35], and miR-30b-3p in glioma [36]. The expression of miR-21 microRNA has been found to be altered in almost all types of cancers, and it has been classified as an oncogenic microRNA [37]. Due to the critical functions of its target genes in various signaling pathways, miR-21 is overexpressed in almost all cancer types, where its upregulation promotes cell proliferation, invasion, and metastatic behavior. Furthermore, miR-21 targets several genes such as RECK, which are involved in metastatic pathways in several tumors.

The expression of miR-21 is highly upregulated in tumor and non-tumor tissues, and it has been linked to low-level expression of RECK. For instance, it has been reported that the overexpression of miR-21 negatively regulates RECK gene expression in glioma [38,39], nerve injury [40], osteosarcoma [41], esophageal squamous cell carcinoma [42], and colon cancer [43]. Previous studies have reported that miR-21 may function as an oncogene in various human cancers [44,45,46,47], including gynecological cancers [48,49]. Recently, several significant miR-21 targets associated with malignancy have been identified by different groups: phosphatase and tensin homolog deleted on chromosome ten (PTEN) [50], programmed cell death 4 protein (PDCD4) [51], maspin [52], tropomyosin 1 (TPM1) [53], acetyl-CoA acetyltransferase 1 (ACAT1) [54], and cyclin-dependent kinase 6 (CDK6) [55]. In particular, the molecular mechanism through which the RECK gene is regulated by miR-21 in cervical cancer is poorly studied. An event that occurs in HPV-associated carcinogenesis during HPV DNA integration is a global perturbation of cellular gene expression [56]. Recent evidence suggests a relationship between HPV E6 and E7 oncogene expression and disruption of cellular microRNA expression. Therefore, it is plausible that HPV infection causes aberrant cellular gene expression, including disruption of microRNA expression.

In the present study, CaSki and SiHa cells (both cervical cancer cells HPV16+) were used to investigate whether siRNA-mediated gene silencing specific to miR-21 expressed in plasmids could alter the expression of the RECK human gene. To fulfill this purpose, we generated siRNA expression plasmids for miR-21. We identified that siRNAs against miR-21 induced the reestablishment of RECK gene and protein expression, as well as the reestablishment of its biological effects on cell proliferation and migration. To describe the molecular pathway of RECK gene regulation by miR-21 and test its potential trans-regulatory properties, we analyzed the effect of miR-21 on the RECK 3′-UTR regulatory region in CaSki and SiHa cells. We found that miR-21 can trans-regulate the RECK 3′-UTR. This effect results from miR-21’s interaction with MRE (microRNA response element) specific binding sites, designated as MRE21-1, MRE21-2, and MRE21-3. These findings support the notion that siRNAs directed against miR-21 serve as excellent molecular tools to inhibit this microRNA’s expression and activities in a targeted manner, thereby inducing the reestablishment of target gene expression, which has significant biological effects on the cervical carcinogenesis process.

## 2. Results

### 2.1. miR-21 and RECK Gene Expression in HPV-Transformed Cervical Cancer Cells

Several studies indicate that overexpression of miR-21 [47,57,58] and decreased expression of RECK [14,59,60] in different malignancies are probably involved in the metastasis process. Such evidence supports a tight association between disruption of miR-21 and RECK gene expression and the development of gynecological malignancies. To assess the pattern of miR-21 and RECK gene expression in cervical cancer cells, we first analyzed endogenous miR-21 and RECK mRNA levels in HPV-negative human epidermal primary keratinocytes (HaCaT cells) and in human cervical cancer cells transformed with HPV16 (CaSki and SiHa cells) by real-time RT-qPCR. Consistent with previous observations from other groups, there is a significantly increased expression of miR-21 mRNA in HPV-positive transformed CaSki and SiHa cells compared with HPV-negative non-transformed HaCaT cells (Figure 1). In addition, there is decreased expression of RECK mRNA in CaSki and SiHa cells compared with HaCaT cells. These results suggest that human cervical cancer cells transformed with HPV16 are specifically able to mediate RECK transcriptional activation during the cervical carcinogenesis process associated with HPV transformation.

### 2.2. siRNA-Mediated Silencing of miR-21 Expression Has an Effect on RECK Expression

In exploring miR-21 target genes, we focused on the RECK gene, a tumor suppressor gene whose protein product functions as a negative regulator for MMPs. To achieve this aim, CaSki and SiHa cells were transiently transfected with the pSIMIR21-5P siRNA expression plasmid to induce the silencing of miR-21, which is over-expressed in this type of cells, as previously demonstrated. To determine whether the RECK gene is a cellular target of miR-21 in cervical cancer cells, we analyzed RECK gene expression in CaSki and SiHa cells transfected with pSIMIR21-5P plasmid, using end-point RT-PCR. As shown in Figure 2, we found that siRNA against miR-21 influences the expression of RECK mRNA. Specifically, we found significant reestablishment of RECK mRNA expression when CaSki and SiHa cells were treated with siRNAs to miR-21. After 48 h of transfection with pSIMIR21-5P, the RECK expression level increased compared with cells that were untreated or transfected with pSilencer 1.0-U6 empty vector plasmid. We observed differences in RECK mRNA expression levels in CaSki and SiHa cells transfected with 3 μg and 5 μg pSIMIR21-5P plasmid, respectively, compared with untreated CaSki and SiHa cells. The control, GAPDH, mRNA expression level did not show any changes in similar transfection conditions.

The effect of siRNAs on miR-21 can be influenced by the secondary structure and positioning of the cognate sequence within the human pre-miR-21 molecule. To analyze the effect of pSIMIR21-5P plasmid on RECK gene expression, we investigated whether siRNAs could induce specific alteration of RECK expression after transient transfection of pSIMIR21-5P plasmid. The next step was analyzing this regulation mechanism quantitatively. For this purpose, CaSki and SiHa cells were transiently transfected with the pSIMIR21-5P plasmid, and we analyzed the RECK expression level by real-time RT-qPCR. As shown in Figure 3, there was a significant increase in the RECK transcript level when CaSki and SiHa cells were transfected with pSIMIR21-5P plasmid at increased concentrations. After 48 h of transfection, the RECK expression level increased by 2.5 times in a short period of time (6 h), compared with cells transfected with pSilencer1.0-U6 empty vector plasmid or non-treated cells. We did not observe differences in RECK expression levels between CaSki and SiHa cells treated with pSilencer1.0-U6 empty vector plasmid compared with non-transfected cells. The RNU6 RNA expression level did not show any changes under these same conditions. These data suggest that pSIMIR21-5P is a siRNA expression plasmid specific for miR-21, and that miR-21 can induce selective and specific silencing of the RECK gene in human cervical cancer cells infected with HPV16.

In addition, we analyzed whether the silencing effect of miR-21 alters RECK protein expression. Using a Western blot assay, we identified the reestablishment and overexpression of RECK cellular protein expression after treatment of CaSki and SiHa cells with a high concentration of siRNAs to miR-21 (Figure 4). After 48 h of transfection, the RECK protein expression level increased by 2 times compared with cells transfected with pSilencer1.0-U6 empty vector plasmid or non-treated cells. We did not observe differences in RECK protein expression levels between CaSki and SiHa cells treated with pSilencer1.0-U6 compared with non-transfected cells. We used beta-actin protein as a control and did not observe any alteration in beta-actin cellular protein expression when CaSki and SiHa cells were transfected with the pSIMIR21-5P plasmid. Thus, our results demonstrate that treatment of CaSki and SiHa cells with siRNAs expressed in plasmid specific for miR-21 induces repression of miR-21 and reestablishment of RECK gene expression and its protein product. Thus, the expression of miR-21 microRNA is inversely correlated with RECK expression, suggesting that RECK is a miR-21 target gene in HPV16+ human cervical cancer cells.

### 2.3. Specific MRE Recognition Sequences by miR-21 Are Critical for Regulation of RECK

In an effort to demonstrate that miR-21 directly targets the RECK gene, three independent luciferase reporter plasmids were generated (pMRE21RECKLuc1, pMRE21RECKLuc2, and pMRE21RECKLuc3), containing cloned MREs (microRNA response elements) to miR-21 from the RECK 3′-UTR regulatory region (MRE21-1 of 3343 to 3366 nt, MRE21-2 of 3893 to 3918 nt, and MRE21-3 of 4039 to 4061 nt) (Figure 5A). The annealing was generated from the target scan human and miRTarBase website database [61].

CaSki and SiHa cells were transiently transfected with pMRE21RECKLuc1, pMRE21RECKLuc2, and pMRE21RECKLuc3 reporter plasmids independently to determine the contribution of each MRE21 recognition site and, subsequently, co-transfected with the pSIMIR21-5P plasmid to determine the effect of silencing miR-21. In CaSki cells, we observed that transfection with pMRE21RECKLuc1 induced a luciferase activity of 70%, the transfection with pMRE21RECKLuc2 caused an increase in luciferase activity of approximately 30%, while transfection with pMRE21RECKLuc3 only induced 10% of luciferase activity, compared with cells transfected with positive control pMIR-Report-Luciferase empty vector plasmid. When CaSki cells were co-transfected with pMRE21RECKLuc1 and pSIMIR21-5p plasmids, the luciferase activity was reestablished very similar to cells transfected with the pMIR-Report-Luciferase plasmid, at approximately 100%. When CaSki cells were co-transfected with pMRE21RECKLuc2 and pSIMIR21-5P plasmids, luciferase activity increased to 90% in comparison with transfection with pMIR-Report-Luciferase empty vector plasmid. When CaSki cells were co-transfected with pMRE21RECKLuc3 and pSIMIR21-5P plasmids, luciferase activity was not induced, and we only observed a luciferase activity of 7% in comparison with transfection with pMIR-Report-Luciferase empty vector plasmid (Figure 5B). In SiHa cells, we observed that transfection with pMRE21RECKLuc1, pMRE21RECKLuc2, and pMRE21RECKLuc3 induced a luciferase activity of 10%, compared with cells transfected with positive control pMIR-Report-Luciferase empty vector plasmid. When SiHa cells were co-transfected with pMRE21RECKLuc1 and pSIMIR21-5p plasmids, the luciferase activity was not reestablished. When SiHa cells were co-transfected with pMRE21RECKLuc2 and pSIMIR21-5P plasmids, luciferase activity increased to 110% in comparison with transfection with pMIR-Report-Luciferase empty vector plasmid. When SiHa cells were co-transfected with pMRE21RECKLuc3 and pSIMIR21-5P plasmids, we observed a luciferase activity of 60% in comparison with transfection with pMIR-Report-Luciferase empty vector plasmid (Figure 5B). Furthermore, when CaSki and SiHa cells were transfected with pMRE21RECKLuc1, pMRE21RECKLuc2, and pMRE21RECKLuc3 and co-transfected with pSIMIR21-5P plasmids, we observed a differential expression of the luciferase activity. This differential expression could be influenced by varying lengths of RECK 3′-UTRs, by the sequence context of a nucleotide complementary between the seed sequence of MRE21-1, MRE21-2, or MRE21-3 with RECK 3′-UTR complementary sequence, or by secondary local RNA structure. These data suggest that MRE21-1, MRE21-2, and MRE21-3 sequences are the main recognition sites through which miR-21 mediates the regulation of the RECK gene. MRE21-1 is located from 3343 to 3366 nt, MRE21-2 is located from 3893 to 3918 nt, while MRE21-3 is located from 4039 to 4061 nt, in RECK 3′-UTR sequence downstream of the transcriptional start site of RECK gene. MRE21-1, MRE21-2, and MRE21-3 appear particularly important for miR-21’s targeting mechanism of RECK gene, given that these MRE-binding sequences induced greater regulation of luciferase reporter gene of the differential manner in CaSKi and SiHa cells. However, further investigation is needed to elucidate the specific molecular mechanism by which miR-21 interacts with MREs on the RECK gene. These findings do suggest that a possible mechanism by which miR-21 regulates the RECK gene in human cervical cancer cells is through interaction with the MRE21-1 and MRE21-2 recognition sites.

### 2.4. Silencing of miR-21 MicroRNA by siRNAs Induces Alterations in Tumor Cell Migration

To examine whether silencing of miR-21 gene expression by siRNAs would affect cellular proliferation and viability, initial MTS assays were carried out on days 0, 1, 2, 3, 4, and 5 after transfection, using equal amounts of CaSki cells transfected with pSIMIR21-5P plasmid. Figure 6A,B show the effect on cellular proliferation and viability after no transfection (Figure 6A, NT, panels a–f), transfection with pSilencer 1.0-U6 empty vector plasmid (Figure 6A, panel aa–cf), and transfection with pSIMIR21-5P plasmid (Figure 6A, panel da–ff). We observed that silencing with siRNAs to miR-21 decreased the viability of CaSki cells compared with non-transfected cells or transfected cells with pSilencer 1.0-U6 empty vector plasmid. Interestingly, we identified a marked decrease in cellular viability from days 4 to 5 after transfection with pSIMIR21-5P siRNA expression plasmid. At 48 h post-transfection, proliferation of CaSki cells was decreased by approximately 10% more than non-transfected cells, while proliferation of CaSki cells was reduced by 20% at day 4, and 40% at day 5 post-transfection (Figure 6B).

To evaluate if silencing of miR-21 gene expression induced by siRNAs expressed in plasmids has effects on cellular mobility, functional assays were carried out in order to determine the effect of silencing miR-21 on migration of the CaSki and SiHa cells. We evaluated the effect of silencing of miR-21 on cellular migration through wound-healing assays. The results show that after 24 h of transfection with pSIMIR21-5P plasmid, silencing miR-21 induced decreased migration of CaSki cells (Figure 7A, panels ah and al) compared with non-transfected CaSki cells (Figure 7A, panels a–d and aa–ad) or transfected with pSilencer 1.0-U6 empty vector plasmid (Figure 7A, panels e–l). The number of migrating CaSki cells was significantly lower in those cells transfected with pSIMIR21-5P plasmid than in either those cells transfected with the pSilencer 1.0-U6 plasmid or non-transfected cells. At 12 h and 24 h post-transfection with pSIMIR21-5P plasmid, mobility of CaSki cells was approximately 10%, while in non-transfected cells or transfected with pSilencer 1.0-U6 empty vector plasmid, mobility was approximately 100% (Figure 7B). Furthermore, we evaluated the effect of silencing of miR-21 on cellular migration through wound-healing assays in SiHa cells. The results show that after 24 h of transfection with pSIMIR21-5P plasmid, silencing miR-21 induced decreased migration of SiHa cells (Figure 8A, panels e and f) compared with transfected SiHa cells with pSilencer 1.0-U6 empty vector plasmid (Figure 8A, panels a–d). The number of migrating SiHa cells was significantly lower in those cells transfected with pSIMIR21-5P plasmid than in either those cells transfected with the pSilencer 1.0-U6 plasmid. At 12 h and 24 h post-transfection with pSIMIR21-5P plasmid, mobility of SiHa cells was approximately 3%, while in transfected with pSilencer 1.0-U6 empty vector plasmid, mobility was approximately 100% (Figure 8). Thus, this type of cellular mobility is characteristic of cells that are incapable of lysing the extracellular matrix and moving through spaces surrounding the extracellular matrix. These data suggest a differential migratory phenotype in cervical cancer cells that may be induced by RECK gene overexpression mediated by silencing of miR-21 microRNA. Thus, these findings support the idea that silencing miR-21 helps reestablish tumor suppressor RECK gene expression, altering cell migration in HPV-transformed cervical cancer cells.

## 3. Discussion

Cervical cancer is caused by long-term infection with high-risk oncogenic HPVs. The cervical carcinogenesis process requires the sustained expression of viral E6 and E7 oncogenes, which induce cell immortalization, resistance to apoptosis, evasion of innate and adaptive immune responses, and specifically in the context of metastasis, alter the expression and activity of extracellular matrix components and their inhibitors such as RECK [62]. The major novel contribution of our data is to demonstrate that RECK gene expression is regulated by miR-21 in HPV-positive cervical cancer cells, and this regulation occurs mainly through MRE21-1, MRE21-2, and MRE21-3 within the RECK 3′-UTR regulatory region. Silencing miR-21 induces the reestablishment of RECK gene and protein expression, leading to biological effects which include inhibition of cervical cancer cell migration and proliferation. This evidence strongly supports the notion of a regulatory genetic network between miR-21 and RECK in HPV-transformed cervical cancer cells.

The direct effect of HPV oncogenes on the expression of different extracellular matrix components has been investigated. A previous study demonstrated that c-Jun inhibition in HPV-transformed cell lines (HeLa and CaSki cells) was associated with lower MMP-9 mRNA expression levels [63]. Upregulation of MMP-2 and MMP-9 expression and activity are the most common extracellular matrix-related modifications in precursor cervical lesions and invasive carcinoma [62]. Interestingly, it has been reported that this effect correlates with an imbalance in the expression of MMPs and inhibitors, including RECK gene expression [12,64]. It has been demonstrated that HPV16-E6/E7 oncoproteins are involved in the regulation of gene expression of MMPs (MT1-MMP, MMP-2, and MMP-9) and the migration of CaSki and SiHa cervical cancer cells [65]. It has also been shown that MMPs and their natural inhibitors, such as RECK protein, are important in normal tissue maintenance and remodeling and play a major role in the transformation process [66]. RECK overexpression appears to delay tumor growth and increase overall survival in vivo. Lower RECK mRNA levels are associated with cervical lesion progression and poor response to chemotherapy in cervical cancer patients. Thus, RECK downregulation is a consistent and clinically relevant event in the natural history of cervical cancer [66].

RECK is a serine protease inhibitor that regulates various physiologic and pathologic events. In cervical carcinoma, RECK expression is associated with the recruitment of tumor-infiltrating lymphocytes and the expression of immune checkpoint molecules, including programmed cell death ligand 1 (PD-L1). PD-L1 is more frequently expressed by squamous cell carcinoma than by adenocarcinoma, which accounts for approximately 20–25% of all cervical malignancies. Diffuse PD-L1 expression in patients with squamous cell carcinoma is correlated with poor disease-free survival and disease-specific survival compared with marginal PD-L1 expression, which is associated with a remarkably favorable prognosis. In adenocarcinoma, there is a survival benefit for patients with tumors lacking PD-L1-positive tumor-associated macrophages [67]. In hepatocellular carcinoma, RECK expression is associated with less vascular invasion, as well as PD-L1 expression [68]. In the muscle regeneration process, it has been reported that notch-1 signaling is reduced, while p38 and AKT signaling are augmented in myoblasts with decreased RECK expression levels [69]. Our understanding of the RECK molecular regulation mechanism mediating miR-21 microRNA function in cancer generally and the cervical carcinogenesis process specifically remains limited. A recent study reported that efficient knockdown of miR-21 by hierarchically tumor-activated nanoCRISPR-Cas13a (CHAIN) restored RECK expression and crippled downstream MMP-2, which undermined cancer proliferation, migration, and invasion in a hepatocellular carcinoma mouse model [59].

In the present study, we first demonstrated that levels of miR-21 mRNA expression in HPV-transformed cervical cancer cells were elevated in comparison with HPV-negative non-transformed cells. Interestingly, we identified an inverse correlation of miR-21 and RECK mRNA gene expression in the cells analyzed (Figure 1). Our data are consistent with other groups’ reports that changes in RECK gene expression could represent a risk factor in cervical cancer development. In a previous study, the levels of RECK expression were analyzed in samples from women with high-grade cervical lesions (CIN II/CIN III) and cervical squamous cell carcinoma (SCC) [64]. There was higher expression of MMP-9 and lower expression of RECK in women with CIN II/CIN III/SCC when compared with women from the control group with no neoplasia or CIN I. Our in vitro findings are consistent with earlier studies and support the theory that an important event in cervical cancer progression consists of decreased RECK gene expression, associated with high miR-21 gene expression in HPV-transformed cervical cancer cells.

Several factors have been identified that upregulate miR-21 microRNA expression, among them are signal transducer and transcription factors, which are constitutively activated in a variety of cancer types, including cervical cancer. We previously demonstrated a physical interaction and functional cooperation between AP-1 transcription factor in the miR-21 promoter, which may explain the contributions of AP-1 in the overexpression of miR-21 in cervical cancer cells [70]. Another factor that can induce miR-21 gene expression is hypoxia, a general feature of all solid tumors. In cervical cancer progression, both AP-1, hypoxia-inducible factor 1 alpha (HIF1A), and S100A2, which are major mediators of cell response to hypoxia, are associated with the carcinogenesis process. Moreover, HIF1A and S100A2 are upregulated and can possibly account for miR-21 overexpression to explain in part the aggressive cervical cancer phenotype progression [71].

Herein, we describe the effect of siRNAs to miR-21 expressed in plasmids on RECK gene expression and protein expression in HPV-transformed cervical cancer cells. CaSki and SiHa cells were transiently transfected with the pSIMIR21-5P plasmid, and RECK gene expression was analyzed by endpoint, real-time RT-qPCR, while RECK protein expression was analyzed by Western blot (Figure 2, Figure 3 and Figure 4). We observed that RECK gene expression and RECK cell protein expression were both reestablished in HPV16-transformed cervical cancer cells (CaSki and SiHa cells) when miR-21 expression was silenced with specific siRNAs to miR-21. These data suggest that RECK gene expression is modulated at the post-transcriptional level by miR-21. Furthermore, RECK regulation by miR-21 may be mediated through miR-21’s interaction with MREs localized in the 3′-UTR regulatory region of the human RECK gene (Figure 5).

The role of microRNAs targeting RECK gene expression has been analyzed by several groups. In other malignancies, it has been shown that miR-21 can bind directly to regulate RECK mRNA through the MREs predicted binding sites in the RECK 3′-UTR. Loayza-Puch et al. demonstrated the interaction between miR-21 and an MRE of RECK 3′-UTR region, using the pRL-TK vector reporter assay, in colon carcinoma [17]. In this study, a database search identified a potential target binding site for miR-21. The function of this MRE was analyzed by introducing synthetic miRNA precursors, and antisense oligonucleotides against miR-21, into colon carcinoma cells, which endogenously overexpress miR-21. The effect was examined by analysis of a luciferase reporter gene linked to the MRE in colon carcinoma cells. Results demonstrated that RECK mutants lacking the MRE had augmented tumor/metastasis-suppressor activities and confirmed that miR-21 is involved in the post-transcriptional downregulation of the RECK gene in colon cancer cells.

Gabriely et al. reported RECK gene and protein expression levels in human gliomas [38]. Results obtained by RT-qPCR and Western blot indicate that RECK mRNA and protein expression are significantly lower in glioma cells than in normal brain tissue. Furthermore, the analysis of MRE of miR-21 binding site from RECK 3′-UTR by cloning into pMir-Report plasmid vector was carried out. Data show that the RECK gene is post-transcriptionally regulated by miR-21, while RECK protein levels are upregulated by miR-21 inhibition in glioma cells.

Han et al. identified a positive correlation between the beta-catenin/STAT3/miR-21 circuit and RECK gene regulation and pathological grade in glioma tissues [39]. They demonstrated that the beta-catenin pathway regulates miR-21 expression in human umbilical vein endothelial cells and glioma cells and that this regulation is signal transducer and activator of transcription 3 (STAT3)-dependent. Furthermore, via chromatin immunoprecipitation and luciferase reporter analysis, the authors demonstrated that miR-21 is controlled by an upstream promoter containing a conserved STAT3-binding site. Notably, the knockdown of miR-21 inhibited cell invasion by increasing RECK expression and decreased tumor growth in a xenograft model. They demonstrated that RECK is regulated by MRE from RECK-3′-UTR mediated by the beta-catenin/STAT3/miR-21 circuit in glioma cells.

Lin et al. reported that lncRNA GAS5 and RECK are under-expressed, while miR-21 is overexpressed in esophageal squamous carcinoma cells [42]. The prediction and analysis of the binding site of lncRNA GAS5 and miR-21 were performed with a bioinformatics website and were verified by dual luciferase reporter gene assay. The results revealed that GAS5 is bound to miR-21. Dual luciferase reporter gene assays showed that luciferase activity of wild-type-GAS5/miR-21 mimics diminished, suggesting that miR-21 might specifically bind to GAS5. When cells were co-transfected with MRE of RECK 3′-UTR and miR-21 mimics, the relative luciferase activity declined; on the other hand, there was no change in luciferase activity after cells were transfected with MRE muted and miR-21 mimics, indicating that RECK is a direct target gene of miR-21. Furthermore, GAS5 elevation and miR-21 inhibition reduced viability and colony formation ability and enhanced apoptosis of esophageal squamous carcinoma cells after radiation therapy, thus enhancing cell radiosensitivity in esophageal squamous carcinoma cells.

Jung et al. demonstrated that miR-21 regulates the RECK gene in oral cancer cells [72]. Results indicate that RECK is under-expressed, while miR-21 is overexpressed in head and neck cancer cells compared with the normal tissues. An MRE of RECK 3′-UTR and a mutated version were examined by luciferase reporter analyses in head and neck cancer cells. Results demonstrated that transfection of miR-21-mimics reduced the expression of RECK through direct miRNA-mediated regulation, and this microRNA was inversely correlated with RECK in orthotopic xenograft tumors. The data show that the RECK gene is post-transcriptionally downregulated by miR-21 in head and neck cancer cells. Moreover, the keratinization process is associated with poor prognosis of patients with oral cancer, and keratinization-associated microRNAs mediate deregulation of RECK, which may contribute to the aggressiveness of tumors.

Zhang et al. reported that expression of miR-21 is increased in gastric tumor tissues and cancer cell lines compared with normal controls. Moreover, the upregulation of miR-21 is related to H. pylori infection [73]. They performed a luciferase reporter assay and observed a significant decrease in luciferase activity in the presence of MRE of RECK 3′-UTR cloned in pMSCVpuro-miR-21 in gastric tumor cells compared with the controls. To validate whether RECK is a direct target of miR-21, a mutated MRE miR-21-binding site was analyzed in the reporter plasmid, and a loss of repression was observed. These findings indicate that RECK is a direct target of miR-21 in gastric cancer cells.

Hu et al. demonstrated that miR-21 is highly expressed in the subluminal stromal cells at mouse embryo implantation sites in pregnancy [74]. Using luciferase transfection assays with the recombinant vector bearing the MRE miR-21-binding site of the RECK 3-UTR region, they found that luciferase activity was down-regulated by the miR-21 precursor and up-regulated by the miR-21 inhibitor. The data indicate that the RECK gene is a target and is post-transcriptionally regulated by miR-21 during embryo implantation in a mouse model.

Xu et al. reported that RECK is a potential target gene of miR-21 in human non-small cell lung cancer cells [75]. They used a pGCMV/EGFP-hsa-miR-21 interference plasmid to silence miR-21 expression, as well as the matching MRE sequences from RECK 3-UTR region target for miR-21. Results show that miR-21 expression level was significantly reduced by interference plasmid, and RECK gene is a target for miR-21 throughout seed matching sequence of MRE RECK 3′-UTR. These findings reveal that downregulation of miR-21 suppresses the proliferation of non-small-cell lung cancer cells, suggesting that miR-21 is a direct regulator of RECK, which could be the key factor involved in cell proliferation in non-small-cell lung cancer cells.

Fan et al. reported that patients with non-calcified coronary artery lesions and calcified plaques have significantly higher miR-21 compared to normal patients [76]. Results indicate that overexpressing miR-21 induces the expression and secretion of pro-MMP-9 and active-MMP-9 in human macrophages by targeting the RECK gene, and knocking down RECK expression with specific siRNA has similar effects to miR-21 overexpression. Results of reporter plasmid assays indicate that miR-21 directly interacts with the MRE in the RECK 3′-UTR and inhibits its expression through the miR-21 “seed” sequence. The findings suggest that miR-21 downregulates RECK expression by means of inhibiting translation, rather than degradation of target mRNA in patients with coronary atherosclerosis.

Chen et al. reported that group box 1 promotes hepatocellular carcinoma progression through miR-21-mediated matrix metalloproteinase activity [77]. They show that RECK is repressed in tumor tissue compared to matched background liver controls, and the abundance of miR-21 and high mobility group box-1 is inversely correlated with levels of RECK. To confirm the validity of the predicted targets, the 3′-UTR containing the MRE miR-21-binding site of the RECK gene was cloned into pLUC expression vectors. Cells showed basal repression of the luciferase activity due to the presence of endogenous miR-21, while the addition of anti-miR-21 reversed this repression. Luciferase activity was inhibited when vectors with the normal target sequence were treated with pre-miR-21. These findings confirm that RECK translational repression is mediated by the MRE-specific binding of miR-21 in target sequences within the 3′-UTR of the RECK gene. These observations demonstrate that endogenous levels of miR-21 can repress the expression of RECK, and that this repression is reversed with the inhibition of miR-21 in hepatocellular carcinoma cells.

In the studies described above, RECK was reported to be a target gene of miR-21 microRNA. The MRE from RECK 3′-UTR that was analyzed with the experimental strategies described corresponds to an MRE reported in our study. When we performed a deep analysis of this MRE-binding sequence from RECK 3′-UTR, the analysis revealed that the MRE of the miR-21 recognition site corresponds to an MRE that we identified with bioinformatics and denominated as MRE21-3. This finding is relevant because we investigated in our study the effect of miR-21 on the MRE recognition sequences present in RECK 3′-UTR regulatory region in cervical tumor cells. We reported that the regulation mechanism is the result of miR-21’s interaction with MIR21-1 located in positions 3343 to 3366 nt, with MRE21-2 located in positions 3893 to 3918 nt, and with MRE21-3 located in positions 4039 to 4061 nt, respectively. Interestingly, when we analyzed the MRE21-3 located in the positions 4039 to 4061 nt, which was cloned in pMRE21RECKLuc3 plasmid, we found one significantly reduced luciferase activity by approximately 90% when miR-21 was silenced in HPV-transformed CasKi and SiHa cells. However, this luciferase activity was only reestablished in SiHa cells when siRNAs to miR-21 were administered (Figure 5). Thus, this finding supports the notion that gene regulation by microRNAs may be sequence-specific through different MRE recognition sites in each pathology. Determining that MRE-binding sites from the RECK 3-UTR sequence belong to miR-21, we hypothesize that these MRE recognition sites are involved in the regulation of RECK gene expression in cervical cancer cells. The results reported here support the notion that MREs of miR-21 in the RECK gene interact in different ways with miR-21 microRNA. Our experimental data suggest that downregulation of RECK by miR-21 may be selectively regulated in different types of tumor cells. In this study, we demonstrated the effects of MRE21-1, MRE21-2, and MRE21-3 recognition sites on the downregulation of RECK gene expression mediated by miR-21 in human cervical tumor cells.

We also sought to determine whether the silencing of miR-21 by siRNAs has an effect on cancer cell proliferation. Previously we demonstrated that siRNAs can silence miR-21 gene expression, causing a decrease in the level of miR-21 in human cervical cancer cells [50]. To elucidate the mechanism by which the silencing of miR21 inhibits the growth of CaSki cells, we performed cell viability (Figure 6) and cell migration assessments and observed inhibition of proliferation and migration in CaSki (Figure 7) and SiHa cells (Figure 8). Figure 6B shows similar results of proliferation cells when CaSki cells were transfected with 3 µg of pSilencer 1.0-U6 and PSIMIR21-5P, while transfection with 5 µg of PSIMIR21-5P, we observed a measurable effect of proliferation cell. This discrepancy in the proliferation cell could be explained in part by endogenous miR-21 overexpression, which is approximately 3 times higher in CaSki cells than in HaCaT cells (Figure 1A). The upregulation of miR-21 in invasive cervical cancer tissues has been reported, and studies have confirmed that miR-21 promotes proliferation, migration, and invasion in CaSki (HPV16-positive) or HeLa cells (HPV18-positive), by downregulating the expression of tumor-repressive PTEN, RECK, and bcl-2 [75,78]. These effects can be explained by the fact that RECK protein is involved in the suppression of tumor invasion, angiogenesis, and metastasis. This effect is associated with RECK’s function as a specific inhibitor of MMP-2, MMP-9, MT1-MMP, and extracellular MMPs, and glycerophosphodiester phosphodiesterase 2 (GDE2), which stimulates A disintegrin metalloproteinase domain-containing protein 10 (ADAM10 APP) cleavage by shedding and inactivating RECK inhibitor of ADAM10 [79]. A recent study showed that RECK overexpression in cervical cancer-derived cell lines (C33A HPV-negative and HeLa HPV18-positive) was associated with reduced migration and invasion potential [80]. Our data are in accordance with previous reports that RECK overexpression reduced the tumorigenic capacity of cervical cancer cells in vivo [66]. We report that overexpression of miR-21 post-transcriptionally downregulates the expression of RECK and inhibits cell proliferation and migration of human cervical cancer cells transformed with HPV16 (CaSki and SiHa cells). Taken together, these observations indicate that loss or reduction in RECK expression is a common trait in several types of human tumors. This event is often associated with a worse prognosis and increased metastasis.

## 4. Materials and Methods

### 4.1. Cell Lines and Culture Conditions

Human cervical cancer cells transformed with HPV16+ (CaSki and SiHa cells) and human epidermal primary keratinocytes HPV- (HaCaT cells) were obtained from the American Type Culture Collection (ATCC). The cell lines were cultured in Dulbecco’s modified Eagle’s medium (DMEM) (Invitrogen, Carlsbad, CA, USA) supplemented with 10% fetal bovine serum (FBS), penicillin/streptomycin (50 μg/mL), 2 mM L-glutamine, 250 ng/mL fungizone, and maintained in a humidified chamber with an atmosphere of 5% CO_2_ at 37 °C. The total RNA isolation was carried out with TriPure isolation reagent (Roche, Indianapolis, IN, USA) for the end-point RT-PCR and real-time RT-qPCR assays. The cellular protein isolation was performed, and protein concentration was determined by the BCA protein kit (Pierce, Rockford, IL, USA) for the Western blot assays. In addition, CaSki and SiHa cells were used in transfection assays and were analyzed for the luciferase activity assays and cell migration assays.

### 4.2. Transfection Assays with siRNA Expression Plasmids to miR-21

Previously, we generated and demonstrated that pSIMIR21-5P is a siRNA expression plasmid specific for miR-21, which has the ability to induce selective and specific silencing of miR-21 microRNA in human cervical cancer cells infected with HPV16 [50,81]. Thus, CaSki and SiHa cells were transiently transfected with pSIMIR21-5P plasmid to silence miR-21, using polyethylenimine linear MW 25,000 transfection grade (PEI 25K) reagent (Polysciences, Warrington, PA, USA), according to the manufacturer’s instructions. Briefly, one day before the transfection assay, the cells were plated at a density of 1X105 cells per well in a six-well plate containing 2 mL of DMEM with 10% FBS and penicillin/streptomycin. At the time of transfection, the plasmids and PEI reagent were diluted in DMEM and incubated for 30 min at room temperature. The plasmid DNA concentration and PEI reagent were normalized by transfection with pGFP plasmid, and all assays were carried out with 0 μg, 3 μg, and 5 μg of plasmids. CaSki and SiHa cells were incubated with plasmids and PEI reagent for 4 h with DMEM containing 0.5% FBS, rinsed, and replenished with DMEM containing 10% FBS. The plasmids were isolated with PureYield plasmid midiprep system (Promega, Madison, WI, USA), and integrity was verified by DNA sequencing. After 48 h of transfection, the cells were harvested, and RNA isolation was carried out for semiquantitative end-point RT-PCR as well as for quantitative real-time RT-PCR assays. Cellular protein isolation was performed by Western blot assays. After transfection, evaluation of reporter plasmid activity for the luciferase activity assays and cell migration assays was performed. Transfection assays were repeated at least four times independently.

### 4.3. Semiquantitative End-Point RT-PCR Analysis of RECK Gene

Transfected CaSki and SiHa cells were harvested and processed for total RNA isolation using TriPure isolation reagent (Roche, Indianapolis, IN, USA) according to the manufacturer’s protocol. Briefly, cells were washed with 1X PBS, and 1 mL TriPure was added. About 200 μL chloroform was added, and the cells were centrifuged. The aqueous phase was separated, and the RNA was precipitated with isopropanol. The RNA was dissolved in DEPC water and the concentration was measured. The mRNA was obtained using oligo dT dT15-18 (Promega, Madison, WI, USA), and cDNA synthesis was performed by incubation with M-MLV reverse transcriptase (Promega, Madison, WI, USA) at 37 °C for 1 h. Homo sapiens RECK gene expression [NCBI: NM_021111.3] was measured by semiquantitative end-point RT-PCR using the sense 5′-TAT-CCA-ATG-AGG-AAC-CCA-AC-3′ and antisense 5′-TTT-AGT-GCA-GAG-TTC-CCT-AC-3′ primers, which were generated using the primer quest tool PCR primer design software from IDT https://www.idtdna.com/pages/tools/primerquest?returnurl=%2Fprimerquest%2Fhome%2Findex (accessed on 22 April 2022) [82]. The PCR reaction amplification conditions were 95 °C for 10 min, 95 °C for 1 min, 55 °C for 30 s, and 72 °C for 1 min for 35 cycles followed by 72 °C for 10 min. A 309 bp DNA fragment was obtained for the RECK gene. Homo sapiens glyceraldehyde-3-phosphate dehydrogenase (GAPDH) [NCBI: NM_002046.6] housekeeping gene was used as a control for gene expression using sense 5′-CAA-CAG-CCT-CAA-GAT-CAT-C-3′ and antisense 5′-ACC-AGG-AAA-TGA-GCT-TGA-C-3′ primers, which were generated using the software from IDT https://www.idtdna.com/pages/tools/primerquest?returnurl=%2Fprimerquest%2Fhome%2Findex (accessed on 22 April 2022) [82]. The PCR reaction amplification conditions were 95 °C for 10 min, 94 °C for 1 min, 54 °C for 1 min, and 72 °C for 1 min for 35 cycles followed by 72 °C for 10 min. A 520 bp DNA fragment was obtained. For each PCR reaction, 1 μg cDNA, 2.5 mM dNTPs, 20 pM each primer, and 0.5 U Taq polymerase (Promega, Madison, WI, USA) were used in a 25 μL volume. To ensure that amplification remained within the linear range, 1:5 serial dilutions of cDNA were made. PCR product bands were digitalized and analyzed by densitometer using ImageJ, which is a Java-based image-processing program. ImageJ program can display, edit, analyze, process, save, and print 8-bit color and grayscale, 16-bit integer, and 32-bit floating point images. It can read many image file formats, including TIFF, PNG, GIF, JPEG, BMP, DICOM, and FITS, as well as raw formats.

### 4.4. Quantitative Real-Time RT-PCR Analysis or RECK and miR-21 Genes

Total RNA isolation from CaSki and SiHa cells transfected as previously described was carried out with TriPure isolation reagent (Roche, Indianapolis, IN, USA) at 0 h, 6 h, 12 h, and 24 h post-transfection. The QuantiNova sybr Green PCR Kit (Qiagen, Hilden, Germany) was used for RT-qPCR assays. The PCR reaction started with an initial incubation step at 95 °C for 2 min to activate the QuantiNova DNA polymerase. The two-step PCR cycling protocol, which has a denaturation step at 95 °C for 5 s and a combined annealing/extension step at 60 °C for 10 s, was standardized for primers with a Tm well below 60 °C. *Homo sapiens* RECK gene expression was measured by real-time RT-qPCR using the sense 5′-ACA-CTA-ATC-CAG-GTG-CCA-TC-3′ and antisense 5′-TTT-CTA-AGA-GTC-CAC-TTT-GTC-C-3′ primers, which were generated using the software from IDT https://www.idtdna.com/pages/tools/primerquest?returnurl=%2Fprimerquest%2Fhome%2Findex (accessed on 22 April 2022) [82]. *Homo sapiens* GAPDH gene expression was used as a control using sense 5′-CAG-GGC-TGC-TTT-TAA-CTC-TGG-TAA-3′ and antisense 5′-GGG-TGG-AAT-CAT-ATT-GGA-ACA-TGT-3′ primers, which were generated using the software from IDT https://www.idtdna.com/pages/tools/primerquest?returnurl=%2Fprimerquest%2Fhome%2Findex (accessed on 22 April 2022) [82]. The reaction was incubated in a 96-well plate at 95 °C for 2 min to PCR initial heat activation, 95 °C for 5 s to denaturation, and 60 °C for 10 s to combined annealing/extension in 2-step cycling for 40 cycles, in a Bio-Rad CFX96 real-time PCR instrument. Human hsa-mir-21 gene expression [miRbase: MI0000077] was measured by real-time RT-qPCR using the sense 5′-ACA-CTC-CAG-CTG-GGT-AGC-TTA-TCA-GAC-TGA-3′, antisense 5′-GTG-TCG-TGG-AGT-CGG-CAA-TTC-3′, and stem-loop 5′-CTC-AAC-TGG-TGT-CGT-GGA-GTC-GGC-AAT-TCA-GTT-GAG-TCA-ACA-TC-3′ primers, which were previously reported [83]. The reaction was incubated in a 96-well plate at 50 °C for 10 min to loop primer integration and cDNA generation, 95 °C for 2 min to denaturation, 95 °C for 5 s to denaturalization, 58 °C for 30 s to annealing, and 60 °C for 10 s to extension for 40 cycles in a Bio-Rad CFX96 real-time PCR instrument. The RT-qPCR reaction was performed by incubation with 50 ng RNA, 5 µL of 2X sybr green PCR master mix, 0.1 µL of QN rox reference dye, and 10 pM of sense and antisense primers in a one-step 10 µL volume reaction. For all PCR reactions, we performed a melting curve analysis to verify the specificity and identity of PCR products. The Ct values were analyzed to determine the statistical significance of RECK and miR-21 gene expression in CaSki cells transfected or non-transfected with pSIMIR21-5P expression plasmid. Relative expression was calculated using the 2^−∆∆Ct^ method and normalized to the expression of GAPDH to RECK and RNU6 to miR-21 [84,85]. All RT-qPCR reactions were performed in triplicate.

### 4.5. Western Blot Assays

CaSki and SiHa cells were harvested, and protein was isolated for Western blot assays 48 h after transfection assays. Briefly, the cells were washed with 1X PBS and incubated for 30 min at 4 °C with lysis buffer containing 50 mM Tris-HCl, 150 mM NaCl, 0.5% SDS, 1% NP40, 0.5 mM AEBSF, 10 μg/μL antipain, 10 μg/μL aprotinin, 10 μg/μL chymostatin, 10 μg/μL leupeptin, 10 μg/μL pepstatin, 1 mM EDTA, 100 mM PMSF, and 0.5 mM DTT (Sigma-Aldrich, Rahway, NJ, USA). The lysates were centrifuged at 11,000 rpm for 15 min. Concentrations of proteins from supernatants were determined using the BCA kit (Pierce, Rockford, IL, USA). Fifty micrograms of protein were electrophoresed on 12% SDS-PAGE, transferred into nitrocellulose membranes, and incubated for antibody detection. Biotinylated and pre-stained molecular weight marker was included. IgG mouse monoclonal antibody RECK (G-4 HRP sc-373929 HRP) was used to detect human RECK protein. Human beta-actin protein was detected using IgG polyclonal antibody (sc-1616-HRP) (Santa Cruz, Biotechnology, Santa Cruz, CA, USA). After the peroxidase-coupled secondary goat antibody mouse anti-IgG was added, bound antibodies and protein were detected by enhanced chemiluminescence using the Renaissance Western blot kit (Pierce, Rockford, IL, USA). The membranes were subjected to autoradiography with an intensifier screen. Western blot assays were performed in triplicate.

### 4.6. Reporter Plasmids and Luciferase Activity Assays

CaSki and SiHa cells were transiently transfected with pMRE21RECKLuc1 plasmid which contains the cloned MRE21-1 (microRNA response element 1 for miR-21 located from 3343 to 3366 nt), with pMRE21RECKLuc2 plasmid which contains the cloned MRE21-2 (microRNA response element 2 for miR-21 located from 3893 to 3918 nt), and with pMRE21RECKLuc3 plasmid which contains the cloned MRE21-3 (microRNA response element 3 for miR-21 located from 4039 to 4061 nt) of RECK 3′-UTR regulatory region. The information was generated from the nucleotide sequence database for RECK [NCBI: NM_021111.3] and for hsa-miR-21 [miRbase: MI0000077]. The RECK 3′-UTR regulatory region sequence downstream of the transcriptional start site of the RECK gene is located at 1410 nt (from 2917 nt to 4326 nt). The MRE21-1, MRE21-2, and MRE21-3 were cloned in Spe I and Hind III restriction sites of pMIR-Report-Luciferase reporter vector plasmid (Life Technologies, Grand Island, NY, USA), which contains a firefly luciferase reporter gene under the control of a CMV promoter/termination system. The design of construct plasmids was carried out in target scan human prediction of microRNA targets software TargetScan Release 7.1 [86]. The pMRE21RECKLuc1 plasmid was generated for cloning of DNA fragment of 53 bp using the sense 5′-CTA-GAC-TAG-TAG-ATA-ATT-ACC-TAC-TCT-GGC-TAG-AAG-CTA-GGG-GTA-AGC-TTG-TG-3′ and antisense 5′-CAC-AAG-CTT-ACC-CCT-AGC-TTC-TAG-CCA-GAG-TAG-GTA-ATT-ATC-TAC-TAG-TCT-AG-3′ primers. The pMRE21RECKLuc2 plasmid was generated for cloning of DNA fragment of 55 bp using the sense 5′-CTA-GAC-TAG-TTT-TTT-GTG-ATA-TGC-ACA-ATG-TAG-ATA-AGT-GTT-CTG-TAA-GCT-TGT-G-3′ and antisense 5′-CAC-AAG-CTT-ACA-GAA-CAC-TTA-TCT-ACA-TTG-TGC-ATA-TCA-CAA-AAA-ACT-AGT-CTA-G-3′ primers. The pMRE21RECKLuc3 plasmid was generated for cloning of DNA fragment of 52 bp using the sense 5′-CTA-GAC-TAG-TAA-TGT-GTT-TCA-CAG-TTT-GAA-ATA-AGC-TAT-TTG-AAA-GCT-TGT-G-3′ and antisense 5′-CAC-AAG-CTT-TCA-AAT-AGC-TTA-TTT-CAA-ACT-GTG-AAA-CAC-ATT-ACT-AGT-CTA-G-3′ primers. The plasmids were isolated by the PureYield plasmid midiprep system (Promega, Madison, WI, USA), and the integrity was verified by DNA sequencing. The co-transfection assays were performed with pSIMIR21-5P plasmid that expresses siRNAs for miR-21. CaSki and SiHa cells were transfected using PEI 25K reagent according to the manufacturer’s instructions as mentioned above. The beta-galactosidase activity was not affected by pMRE21RECKLuc plasmids in cells; therefore, the luciferase activity in all assays was normalized using the pMIR-Report-Luciferase empty vector and pMIR-Report-beta-gal reporter plasmids. Transfection assays were performed with 0 µg, 3 µg, and 5 µg of plasmid DNA. CaSki and SiHa cells were incubated with PEI 25K reagent for 4 h, and 48 h after transfection, cells were washed with 1X PBS and were harvested and lysed with luciferase assay system with reporter lysis buffer (Promega, Madison WI). The cellular extracts were collected by centrifugation, and 50 µg of total proteins was used to determine luciferase activity. Luciferase activity was measured and normalized using the luciferase assay system with reporter lysis buffer (Promega, Madison WI, USA) in Glomax multidetection equipment according to the manufacturer’s instructions. The luminescence was calculated to normalize results with respect to pMIR-Report-Luciferase empty vector plasmid and the efficiency of transfection. All transfections and co-transfections were repeated at least three times independently.

### 4.7. Cellular Viability Assays

Cellular viability was measured using [3-(4,5-dimethylthiazol-2-yl)-5-(3-carboxymethoxyphenyl)-2-(4-sulfophenyl)-2H-tetrazolium] inner salt MTS assay (Promega, Madison, WI, USA), which is a colorimetric method for determining the number of viable cells in a proliferation or cytotoxicity assay. Briefly, a total of 2 × 10^4^ CaSki and SiHa cells per well were plated in a 96-well plate. After 24 h of plating, 20 μL of MTS reagent was added into each well containing the untreated cells and cells transfected with pSilencer 1-0-U6 empty vector plasmid and pSIMIR21-5P plasmid in 100 μL DMEM, and these were incubated at 37 °C for 4 h. MTS tetrazolium compound salt reagent is bioreduced by living cells into a colored formazan product that is soluble in tissue culture medium. After incubation, the absorbance values were measured at 490 nm in Glomax multidetection equipment according to the manufacturer’s instructions. The luminescence was calculated to normalize results with respect to pSilencer 1.0-U6 empty vector plasmid and the efficiency of transfection. The cellular viability rate was calculated as the percentage of MTS adsorption as follows: % survival = (mean experimental absorbance/mean control absorbance) × 100. All transfections and co-transfections were repeated at least three times independently.

### 4.8. Cellular Migration Assays

The capacity for cellular migration was evaluated via wound healing assay. CaSki and SiHa cells were untreated or transiently transfected with 5 µg of pSilencer 1.0-U6 empty vector plasmid and pSIMIR21-5P plasmid. After transfection, cells were cultivated in six-well plates in DMEM with 10% FBS until reaching a confluence of 100%. With the tip of a 200 μL micropipette, a stria was scraped onto the cellular monolayer, and the cultures were washed twice with PBS 1X to remove the displaced or damaged cells and incubated at 37 °C in DMEM with 0.5% FBS for 48 h. The cells that migrated to the center of the wound were documented via microphotographs for 0 h, 6 h, 12 h, and 24 h after the stria had been made. The assays were undertaken in three independent replicates.

### 4.9. Statistical Analysis

All experiments were performed at least three times. The data were analyzed using a student’s *t*-test to identify statistically significant basal gene expression. ANOVA One-Way and Tukey tests were used to perform comparisons between different groups. *p* values less than 0.05 were considered statistically significant and were indicated with an asterisk (*).

## 5. Conclusions

In the present study, we identified the upregulation of miR-21 and determined that it is involved in the proliferation and migration of cervical cancer cells. The principal novel contribution of our data is to demonstrate that RECK gene expression is regulated by miR-21. We confirm that this regulation occurs mainly through MRE21-1, MRE21-2, and MRE21-3 within the RECK 3′-UTR. These results suggest a role of aberrant miR-21 upregulation and an underlying molecular mechanism in cervical cancer cells. Furthermore, this evidence strongly supports the notion of a regulatory genetic network between miR-21 microRNA and the RECK gene. This study shows that silencing of miR-21 induces the reestablishment of RECK gene and protein expression in cervical cancer cells. The reestablishment of RECK may explain the biological effects of silencing miR-21, which include inhibition of cell migration in vitro. This may be due in part to RECK’s function as a specific inhibitor of MMPs. After demonstrating the biological effects of miR-21 on cervical cancer cell growth and migration, we identified a functional target through bio-informatic analysis and luciferase reporter assays. The results suggest that miR-21 can bind to the 3′ UTR of RECK and consequently downregulate its expression. Previous studies have shown that RECK is a membrane-anchored glycoprotein that binds and negatively regulates several MMPs, which proteolytically degrade extracellular matrix proteins, which is critical for tumor metastasis and invasion. The results presented here clearly support the notion that RECK overexpression has a negative effect on the miR-21-mediated tumorigenic potential of HPV-transformed cervical cancer cells. Thus, we report that miR-21 post-transcriptionally downregulates RECK gene expression. Silencing miR-21 inhibits cell migration of HPV-transformed cervical cancer cells.

Our findings suggest that a therapeutic strategy employing siRNAs could effectively inhibit the growth of virus-related cancers. Furthermore, RECK could be a promising prognostic biomarker and may shape a low-metastasis microenvironment, while miR-21 may shape high-tumor progression in patients with cervical cancer. The use of siRNAs has extensive potential to revolutionize every aspect of precision medicine application in biomedical research. Principally, siRNA-mediated innovative advances are increasing rapidly in support of cancer diagnosis and therapeutic purposes. Conversely, it has some delivery challenges to the site of action within the cells of a target organ, due to the progress of nucleic acids engineering and advanced material science research contributing to the exceptional organ-specific targeted therapy. Therefore, siRNA-mediated cancer gene therapy requires attention to each specific cancer by the method of non-invasive siRNA delivery and effective gene silencing approaches. There is no doubt that the siRNA technology platform is a flexible and realistic gene therapy strategy against the development of cervical cancer. The challenge is now to develop efficient strategies for the application of this technology in clinical trials in the context of precision medicine.

An excellent example of this approach is the study reported by Monfared et al., who demonstrated the therapeutic effects of engineered exosomes packed, with miR-21-sponges in glioblastoma [87]. Results revealed that the engineered exosomes have the potential to suppress miR-21 and consequently to upregulate miR-21 target genes, PDCD4 and RECK. In cells treated with miR-21-sponge exosomes, there was a decline in proliferation and also an elevation in apoptotic rates. In a rat model of glioblastoma, administration of exosomes loaded with a miR-21-sponge construct led to a significant reduction in tumor volume. Another example is the study performed by Camargo et al., who reported the effect of gene editing by CRISPR-Cas9 of miR-21 to target MMP9 in metastatic prostate cancer [88]. Their data showed that miR-21 CRISPR-Cas9-edited cells upregulated RECK, MARCKS, BTG2, and PDCD4, while CDH1, ITGB3, and ITGB1 were increased in MMP9 and miR-21 CRISPR-Cas9-edited cells. Reduced cell proliferation, apoptosis, and low invasion in MMP9 and miR-21-edited cells were observed. These findings suggest that CRISPR-Cas9-edited cells for miR-21 and MMP9 impede prostate cancer progression.

As demonstrated in these recent studies, there are several promising technologies for gene therapy in human cancer. In cervical cancer, we propose miR-21 as a target due to the anti-cancer effects of re-establishing RECK.

## Figures and Tables

**Figure 1 ijms-25-04086-f001:**
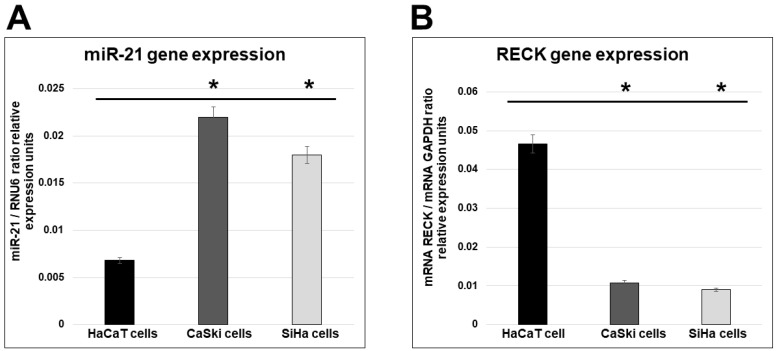
Basal expression analysis of miR-21 and RECK gene expression in HPV-transformed tumor cells. Quantitative real-time RT-PCR analysis of miR-21 (panel **A**) and RECK (panel **B**) gene expression on HaCaT, CaSki, and SiHa cells in basal conditions. Total RNA and cDNA synthesis were obtained from 1 × 10^5^ cells per well in a six-well plate containing DMEM at 37 °C with 5% CO_2_. Relative expression by real-time RT-qPCR analysis of miR-21 was calculated using the 2^−∆Ct^ method and was analyzed by miR-21/mRNA RNAU6 ratio relative expression units. The analysis of RECK was calculated by mRNA RECK/mRNA GAPDH ratio relative expression units. The Ct values were normalized with HaCaT, CaSki, and SiHa cells untreated and are presented as mean ± SD. *p* values < 0.05 are indicated with an asterisk.

**Figure 2 ijms-25-04086-f002:**
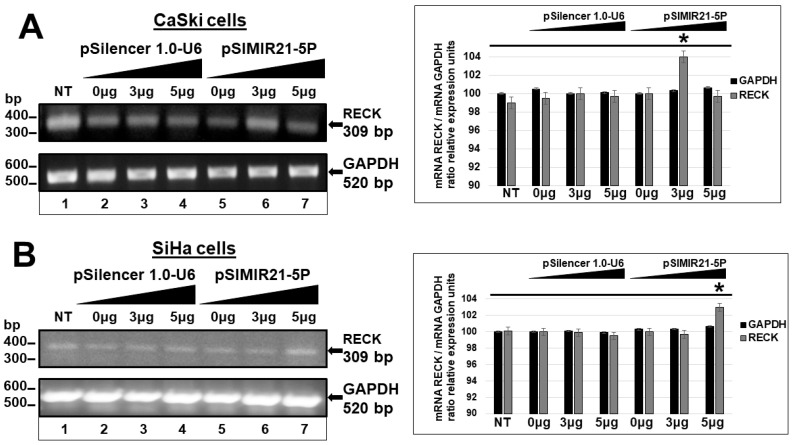
Analysis of RECK gene expression by semiquantitative end-point RT-PCR after miR-21 silencing. Total RNA and cDNA synthesis were obtained from 1 × 10^5^ CaSki and SiHa cells per well in a six-well plate containing DMEM at 37 °C with 5% CO_2_ after 48 h transfection with pSIMIR21-5P plasmid. Panel (**A**): Analysis of RECK gene expression by semiquantitative end-point RT-PCR in CaSki cells that were not transfected (NT, lane 1) or transfected with increased concentration of pSilencer1.0-U6 empty vector plasmid (lanes 2 to 4) or with increased concentration of pSIMIR21-5P plasmid (lanes 5 to 7). PCR amplification products were separated by electrophoresis in 1% agarose gel. The DNA 100 bp ladder was used as molecular weight. Panel (**B**): Analysis of RECK gene expression by semiquantitative end-point RT-PCR in SiHa cells in similar transfection conditions. The graphical representations correspond to PCR product bands, which were digitalized and analyzed by densitometer using the ImageJ program, and data were analyzed by mRNA RECK/mRNA GADPH ratio in relative expression units. The values were normalized with the NT cells (NT) and are presented as mean ± SD. *p* values < 0.05 are indicated with an asterisk.

**Figure 3 ijms-25-04086-f003:**
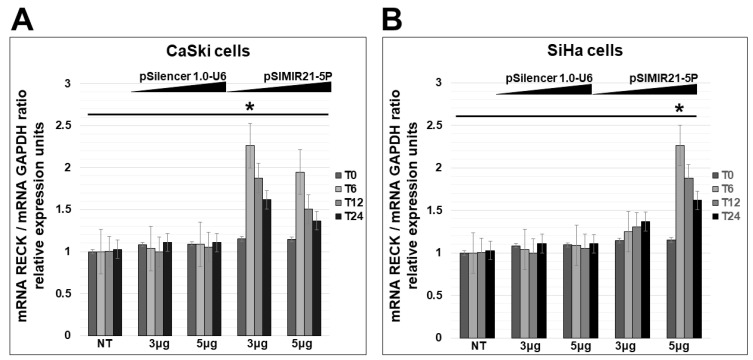
Analysis of RECK gene expression by quantitative real-time RT-PCR after miR-21 silencing. Quantitative real-time RT-PCR analysis of RECK gene expression in CaSki (panel **A**) and SiHa cells (panel **B**) transfected with pSIMIR21-5P plasmid. Total RNA and cDNA synthesis were obtained from 1 × 10^5^ cells per well in a six-well plate containing DMEM at 37 °C with 5% CO_2_ after 0 h, 6 h, 12 h, and 24 h (T0, T6, T12, and T24, respectively) transfection with increased concentrations of pSIMIR21-5P plasmid (0 μg, 3 μg, and 5 μg). Relative expression by real-time RT-qPCR analysis of RECK was calculated using the 2^−∆∆Ct^ method and was analyzed by mRNA RECK/mRNA GAPDH ratio relative expression units. The Ct values were normalized with pSilencer 1.0-U6 empty vector plasmid transfection and are presented as mean ± SD. *p* values < 0.05 are indicated with an asterisk.

**Figure 4 ijms-25-04086-f004:**
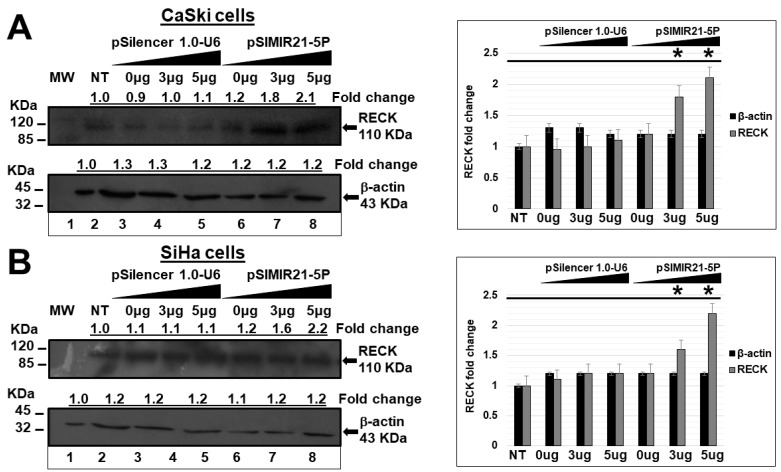
Analysis of RECK protein expression by Western blot after miR-21 silencing. Total cellular proteins were obtained from 1 × 10^5^ CaSki (panel **A**) and SiHa cells (panel **B**) per well in a six-well plate containing DMEM at 37 °C with 5% CO_2_ after 48 h of transfection with pSilencer 1.0-U6 empty vector plasmid or pSIMIR21-5P plasmids (lane 1: MW, molecular weight; lane 2: NT, non-transfected cells; lanes 3 to 5 cells transfected with pSilencer 1.0-U6 plasmid; lanes 6 to 8 cells transfected with pSIMIR21-5P plasmid). The proteins were separated in 12% SDS-PAGE and were transferred to nitrocellulose membranes, which were incubated with each antibody. The RECK and beta-actin protein expression were detected with each antibody, respectively, after transfection with pSilencer 1.0-U6 and pSIMIR21-5P plasmids. Protein amounts were analyzed in the immunoblots. Immunoblot bands were digitalized and analyzed by densitometer using the ImageJ program, and data were analyzed by RECK fold change (mean ± SD). The values were normalized with the NT cells and are presented as mean ± SD. *p* values < 0.05 are indicated with an asterisk.

**Figure 5 ijms-25-04086-f005:**
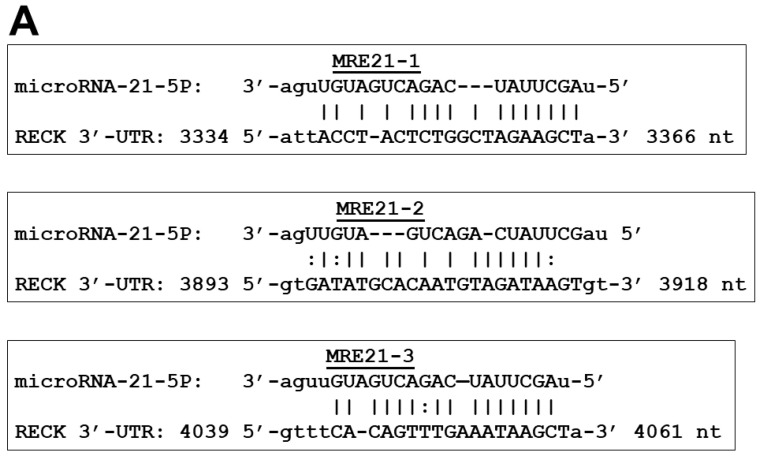
Functional analysis of MRE recognition sequences of RECK gene by miR-21. Panel (**A**): Nucleotide sequences of MRE21-1, MRE21-2, and MRE21-3 of RECK 3′-UTR regulatory region. Nucleotide sequences of microRNAs response elements of miR-21 (MRE21-1 of 3343 to 3366 nt, MRE21-2 of 3893 to 3918 nt, and MRE21-3 of 4039 to 4061 nt) and complementary with RECK 3′-UTR are indicated. Information was generated from the nucleotide sequence database for RECK [NCBI: NM_021111.3] and for hsa-mir-21 [miRbase: MI0000077]. Annealing was generated from the miRTarBase website database. Panel (**B**): Regulation of RECK 3′-UTR region modulated by miR-21 microRNA in CaSki and SiHa cells non-transfected (NT, lane 1) or transfected with pSilencer 1.0-U6 empty vector (lane 2), with pMIR-Report-Luciferase empty vector (lane 3), transfected with pMRE21RECKLuc1 (MRE21-1, lane 4), co-transfected with pMRE21RECKLuc1 and pSIMIR21-5P (lane 5), transfected with pMRE21RECKLuc2 (MRE21-2, lane 6), co-transfected with pMRE21RECKLuc2 and pSIMIR21-5P (lane 7), transfected with pMRE21RECKLuc3 (MRE21-3, lane 8), co-transfected with pMRE21RECKLuc3 and pSIMIR21-5P (lane 9) plasmids. After 48 h of transfection, luciferase activity levels were measured. The data shown represent the average of four independent experiments. *p* values < 0.05 are indicated with an asterisk.

**Figure 6 ijms-25-04086-f006:**
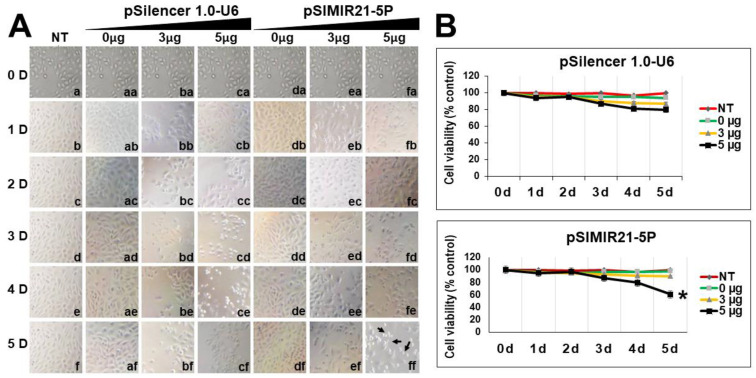
Analysis of tumor cell viability for silencing of miR-21 microRNA expression by siRNAs. Panel (**A**): CaSki cells not transfected (NT, lanes a–f), transfected with pSilencer 1.0-U6 empty vector plasmid (lanes from aa to cf), or transfected with pSIMIR21-5P plasmid (lanes from da to ff) were analyzed by white light microscopy (20×) during 5 days after transfection. Black arrows indicate the dead cells. Panel (**B**): Cellular viability was measured in CaSki cells transfected with pSilencer 1.0-U6 or pSIMIR21-5P plasmids, respectively, using MTS assay. *p* values < 0.05 are indicated with an asterisk.

**Figure 7 ijms-25-04086-f007:**
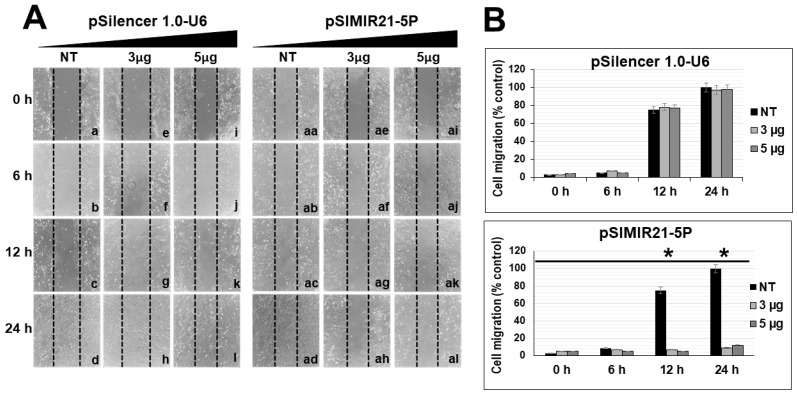
Effect of silencing miR-21 on migration of CaSki cells. Cellular migration was evaluated by a wound healing assay. The microscopy images were captured at 0 h, 6 h, 12 h, and 24 h after the stria was made. The migration of CaSki cells was significantly reduced at 24 h after transfection with siRNAs to miR-21. Panel (**A**): Microscopy images of CaSki cell migration non-transfected (NT, lanes a–d and aa–ad) or transfected with pSilencer 1.0-U6 empty vector plasmid (lanes e–l) or pSIMIR21-5P plasmid, respectively (lanes ae–al). Panel (**B**): Analysis of CaSki cell migration non-transfected or transfected with pSilencer 1.0-U6 or pSIMIR21-5P plasmids. The data are expressed in averages + SE. *p* values < 0.05 are indicated with an asterisk. Following transfection, cells were observed in a Nikon Elipse Ts2 microscopy, and samples were analyzed using the 40× phase contrast objective.

**Figure 8 ijms-25-04086-f008:**
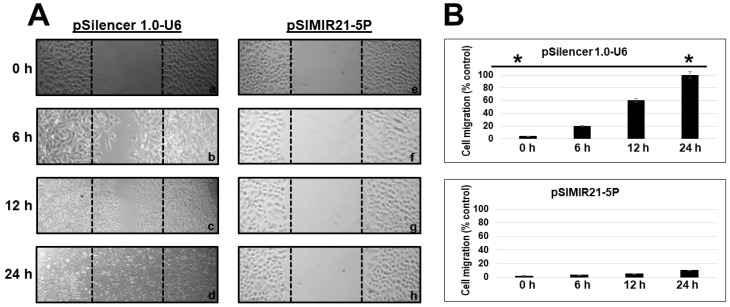
Effect of silencing miR-21 on the migration of SiHa cells. Cellular migration was evaluated by a wound healing assay. The microscopy images were captured at 0 h, 6 h, 12 h, and 24 h after the stria was made. The migration of SiHa cells was significantly reduced at 24 h after transfection with siRNAs to miR-21. Panel (**A**): Microscopy images of SiHa cell migration transfected with pSilencer 1.0-U6 empty vector plasmid (lanes a–d) or pSIMIR21-5P plasmid, respectively (lanes e–h). Panel (**B**): Analysis of SiHa cell migration transfected with pSilencer 1.0-U6 or pSIMIR21-5P plasmids. The data are expressed in averages + SE. *p* values < 0.05 are indicated with an asterisk. Following transfection, cells were observed in a Nikon Elipse Ts2 microscopy, and samples were analyzed using the 40× phase contrast objective.

## Data Availability

Data sharing is not applicable to this article.

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
