# Peer review of "MiR-21 Regulates Growth and Migration of Cervical Cancer Cells by RECK Signaling Pathway"

_ijms, 2024, doi:10.3390/ijms25074086_

Round 1

Reviewer 1 Report

Comments and Suggestions for Authors

Aguilar-Martínez et al. have conducted a study on the role of miR-21, a microRNA commonly overexpressed in various cancers, including cervical cancer, where it acts as an oncogene. Their research focuses on how miR-21 regulates the tumor suppressor gene RECK in cervical cancer cells. By employing siRNAs to inhibit miR-21, they observed an inverse relationship between miR-21 levels and the expression of RECK at both mRNA and protein levels. Furthermore, suppression of miR-21 resulted in increased luciferase activity in constructs containing RECK 3′-UTR microRNA response elements, indicating a direct regulatory action. The study also reveals that reducing miR-21 expression significantly diminishes cervical cancer cell proliferation and migration. These findings suggest that miR-21 downregulates RECK expression to facilitate cancer cell growth and movement, highlighting the potential of targeting miR-21 and RECK in cervical cancer gene therapy.

The claims are properly placed in the context of the previous literature. The experimental data support the claims. The manuscript is written clearly enough that most of it is understandable to non-specialists. The authors have provided adequate proof for their claims, without overselling them. The authors have treated the previous literature fairly. The paper offers enough details of methodology so that the experiments could be reproduced.

Comments

1. The opening sentence of the introduction, stating 'Cervical cancer remains the fourth most frequently diagnosed malignancy in women and one of the leading causes of cancer-related mortality worldwide, especially in low- and middle-income countries,' is frequently used in literature on this topic. To distinguish your manuscript and immediately engage the reader, I suggest starting with a unique introductory sentence that highlights the novel aspects of your study or provides a fresh perspective on cervical cancer's global impact.

2. In the manuscript, the authors state, 'Infection by high-risk oncogenic HPV is associated with the development of pre-cancerous lesions and anogenital carcinomas.' While this accurately reflects the significant impact of HPV on anogenital cancers (including cervical, vaginal, vulval, penile, and anal cancers), it is important to also acknowledge the role of HPV in the etiology of head and neck cancers. The inclusion of information on HPV's association with head and neck cancers would provide a more comprehensive overview of the virus's oncogenic potential across different sites.

3. The manuscript contains unusual word splits, likely a result of earlier text formatting processes. Hyphens appear within words where they typically wouldn't be expected.

For instance, in the phrase 'Transformation of a non-invasive cervi-cal neoplasia into an invasive cervical carcinoma,' the word 'cervical' is arbitrarily hyphenated as 'cervi-cal.' This suggests a need for careful review and correction of these formatting irregularities throughout the document.

Minor revisions

Line 44-46, "Despite advancements in screening and vaccination, cervical cancer continues to pose a significant public health challenge, ranking as a major cause of oncological morbidity and mortality among women globally. This disparity is particularly pronounced in low- and middle-income countries, where access to preventive and therapeutic interventions remains limited."

Line 46-48, "Infection by high-risk oncogenic HPV is associated with the development of precancerous lesions and anogenital carcinomas, currently at the forefront of prophylactic and therapeutic strategy development. Additionally, HPV contributes to head and neck cancers, broadening the spectrum of malignancies linked to this virus and highlighting the necessity for comprehensive management and treatment approaches."

Line 50-53, "Transformation of a non-invasive cervical neoplasia into an invasive cervical carcinoma requires the migration of epithelial cells through the subjacent extracellular matrix, a process in which the matrix metalloproteinases (MMPs) play a critical role by degrading the extracellular matrix components."

Line 114-119, "This effect results from miR-21's interaction with MRE (microRNA response element) specific binding sites, designated as MRE21-1, MRE21-2, and MRE21-3. These findings support the notion that siRNAs directed against miR-21 serve as excellent molecular tools to inhibit this microRNA's expression and activities in a targeted manner, thereby inducing the reestablishment of target gene expression, which has significant biological effects on the cervical carcinogenesis process."

Comments on the Quality of English Language

The English language quality in the provided text is generally good, with clear conveyance of complex scientific information and minimal issues in terms of grammar or syntax. The sentences are well-constructed, and the terminology is used appropriately for a scientific manuscript. There are no major errors that impede understanding, and the text adheres to the conventions expected in scientific writing.

However, to provide a thorough assessment and to be meticulous, there might be a few areas where minor editing could enhance clarity, readability, or conciseness. These potential edits could be in the form of adjusting sentence structure for improved flow, refining the choice of words for precision, or ensuring consistency in tense and voice. Nonetheless, these are relatively minor considerations and do not significantly detract from the overall quality of the English language in the manuscript.

Author Response

Please find enclosed the cover letter to explain, point by point, to the comments raised by the reviewers.

REVIEWER # 1

Point 1. The reviewer said:

  1. The claims are properly placed in the context of the previous literature. The experimental data support the claims. The manuscript is written clearly enough that most of it is understandable to non-specialists. The authors have provided adequate proof for their claims, without overselling them. The authors have treated the previous literature fairly. The paper offers enough details of methodology so that the experiments could be reproduced.

The opening sentence of the introduction, stating 'Cervical cancer remains the fourth most frequently diagnosed malignancy in women and one of the leading causes of cancer-related mortality worldwide, especially in low- and middle-income countries,' is frequently used in literature on this topic. To distinguish your manuscript and immediately engage the reader, I suggest starting with a unique introductory sentence that highlights the novel aspects of your study or provides a fresh perspective on cervical cancer's global impact.

Response to point 1. Thank very much for these comments.

In introduction section, we changed the sentence:

“Cervical cancer remains the fourth most frequently diagnosed malignancy in women and one of the leading causes of cancer-related mortality worldwide, especially in low-and middle-income countries [1]. Infection by high-risk oncogenic HPV is associated with the development of pre-cancerous lesions and anogenital carcinomas, which are currently the focus of attention for the development of prophylactic and therapeutic strategies [2].”

By the sentence:

“Despite advancements in screening and vaccination, cervical cancer continues to pose a significant public health challenge, ranking as a major cause of oncological morbidity and mortality among women globally. This disparity is particularly pronounced in low- and middle-income countries, where access to preventive and therapeutic interventions remains limited [1, 2]. Infection by high-risk oncogenic HPV is associated with the development of precancerous lesions and anogenital carcinomas, currently at the forefront of prophylactic and therapeutic strategy development. Additionally, HPV contributes to head and neck cancers, broadening the spectrum of malignancies linked to this virus and highlighting the necessity for comprehensive management and treatment approaches [3]. Interestingly, National Health Service England first aimed to tackle the burden of the disease by introducing a national cervical screening program in 1988, which has since seen a significant reduction in over a third of cases in England [4]”. Line 44-56.

Point 2. The reviewer said:

  1. In the manuscript, the authors state, 'Infection by high-risk oncogenic HPV is associated with the development of pre-cancerous lesions and anogenital carcinomas.' While this accurately reflects the significant impact of HPV on anogenital cancers (including cervical, vaginal, vulval, penile, and anal cancers), it is important to also acknowledge the role of HPV in the etiology of head and neck cancers. The inclusion of information on HPV's association with head and neck cancers would provide a more comprehensive overview of the virus's oncogenic potential across different sites.

Response to point 2. Thank very much for these comments.

In introduction section, we included this sentence, and we included the reference [3]:

“Infection by high-risk oncogenic HPV is associated with the development of precancerous lesions and anogenital carcinomas, currently at the forefront of prophylactic and therapeutic strategy development. Additionally, HPV contributes to head and neck cancers, broadening the spectrum of malignancies linked to this virus and highlighting the necessity for comprehensive management and treatment approaches [3]”. Line 48-53.

[3]. Leemans, CR.; Snijders, P.J.F.; Brakenhoff, R.H. Publisher Correction: The molecular landscape of head and neck cancer. Nat Rev Cancer. 2018, 18(10), 662. DOI: 10.1038/s41568-018-0057-9. Erratum for: Nat Rev Cancer. 2018, 18(5), 269-282. Line 949-950.

In Introduction section, we included this sentence:

"Transformation of a non-invasive cervical neoplasia into an invasive cervical carcinoma requires the migration of epithelial cells through the subjacent extracellular matrix, a process in which the matrix metalloproteinases (MMPs) play a critical role by degrading the extracellular matrix components". Line 59-63.

In Introduction section, we included this sentence:

"This effect results from miR-21's interaction with MRE (microRNA response element) specific binding sites, designated as MRE21-1, MRE21-2, and MRE21-3. These findings support the notion that siRNAs directed against miR-21 serve as excellent molecular tools to inhibit this microRNA's expression and activities in a targeted manner, thereby inducing the reestablishment of target gene expression, which has significant biological effects on the cervical carcinogenesis process." Line 123-129.

Point 3. The reviewer said:

  1. The manuscript contains unusual word splits, likely a result of earlier text formatting processes. Hyphens appear within words where they typically wouldn't be expected.

For instance, in the phrase 'Transformation of a non-invasive cervi-cal neoplasia into an invasive cervical carcinoma,' the word 'cervical' is arbitrarily hyphenated as 'cervi-cal.' This suggests a need for careful review and correction of these formatting irregularities throughout the document.

Response to point 3. Thank very much for these comments.

We careful reviewed and corrected these formatting irregularities throughout the manuscript. However, the formatting template automatically make the separation of words.

Point 4. The reviewer said:

Comments on the Quality of English Language

The English language quality in the provided text is generally good, with clear conveyance of complex scientific information and minimal issues in terms of grammar or syntax. The sentences are well-constructed, and the terminology is used appropriately for a scientific manuscript. There are no major errors that impede understanding, and the text adheres to the conventions expected in scientific writing.

However, to provide a thorough assessment and to be meticulous, there might be a few areas where minor editing could enhance clarity, readability, or conciseness. These potential edits could be in the form of adjusting sentence structure for improved flow, refining the choice of words for precision, or ensuring consistency in tense and voice. Nonetheless, these are relatively minor considerations and do not significantly detract from the overall quality of the English language in the manuscript.

Response to point 4. Thank very much for these comments.

We reviewed the English language editing of manuscript.

All the comments suggested by the Reviewer were included and are indicated in bold in the manuscript.

Reviewer 2 Report

Comments and Suggestions for Authors

In this article, the authors analyzed the role of miR-21 in RECK gene regulation in cervical cancer cells. They silenced endogenous miR-21 expression using siRNAs to identify the downstream cellular target genes of upstream miR-21. The manuscript is straightforward, well written, and concise and has clear results. Definitely deserves to be published and is a valuable contribution to the “International Journal of Molecular Sciences”. The following comments need to be addressed, as recommended.

[1] “1. Introduction”, Page 1 of 23, Lines 44-46:

“Cervical cancer remains the fourth most frequently diagnosed malignancy in women and one of the leading causes of cancer-related mortality worldwide, especially in low- and middle-income countries [1].”.

Indeed, despite vaccine policies and screening offered by many countries, cervical cancer remains the fourth leading cause of cancer in women worldwide. At that stage, the authors should report that NHS England first aimed to tackle the burden of the disease by introducing a national cervical screening programme in 1988, which has since seen a significant reduction in over a third of cases in England. Cervical cancer screening is available from the age of 25, as the disease is rare among younger individuals.

Recommended reference: Choi S, et al. HPV and Cervical Cancer: A Review of Epidemiology and Screening Uptake in the UK. Pathogens 2023;12(2):298.

[2] “1. Introduction”, Page 2 of 23, Lines 49-50:

Cervical carcinoma has the capacity to metastasize to distant tissues and organs, a fact strongly associated with treatment failure.”.

From the therapeutic perspective, the authors should mention that ongoing immunotherapy trials evaluate vaccine-based therapies, adoptive T-cell therapy and immune-modulating agents in patients with advanced cervical cancer.

[3] 3Discussion”, Page 11 of 23, Lines 381-385:

RECK is a serine protease inhibitor that regulates various physiologic and pathologic events. In hepatocellular carcinoma, RECK expression is associated with less vascular invasion, as well as immunogenic features such as recruitment of tumor-infiltrating lymphocytes and expression of immune checkpoint molecules, including programmed cell death ligand 1 (PD-L1) [64].”.

At that point, the authors should report that PD-L1 is more frequently expressed by squamous cell carcinoma than by adenocarcinoma, which accounts for approximately 20–25% of all cervical malignancies.  Diffuse PD-L1 expression in squamous cell carcinoma patients is correlated with poor disease-free survival and disease-specific survival compared with marginal PD-L1 expression, which is associated with a remarkably favorable prognosis. In adenocarcinoma, there is a survival benefit for patients with tumor lacking PD-L1-positive tumor-associated macrophages.

Comments on the Quality of English Language

Minor editing of English language required

Author Response

Ms. Tamara Ugarković

Assistant Editor, MDPI Belgrade

MDPI Branch Office, Belgrade

Editorial Office

Int J Mol Sci

Dear Ms. Tamara Ugarković

Thank you very much for giving us the opportunity to publish our manuscript entitled: Manuscript ID: ijms-2887481. Type of manuscript: Article: Title: “Mir-21 regulates growth and migration of cervical cancer cells by RECK signaling pathway”. Seidy Y. Aguilar-Martínez, Gabriela E. Campos-Viguri, Selma E. Medina-García, Ricardo J. García-Flores, Jessica Deas, Claudia Gómez-Cerón, Abraham Pedroza-Torres, Elizabeth Bautista-Rodríguez, Gloria Fernández-Tilapa, Mauricio Rodríguez-Dorantes, Carlos Pérez-Plasencia, Oscar Peralta-Zaragoza *, in Int J Mol Sci., in the original articles section.

We are very interested in re-submitting our manuscript and I am attaching a cover letter to explain, point by point, to the comments raised by the reviewers. In addition, I ensured that all changes in the manuscript are indicated in the text in bold by highlighting or using track changes.

I look forward to receiving your positive response in this regard.

Best regards

Oscar Peralta-Zaragoza. Ph. D.

Direction of Chronic Infections and Cancer,

Research Center in Infection Diseases,

National Institute of Public Health.

Av. Universidad No. 655, Cerrada los Pinos

y Caminera. Colonia Santa María Ahuacatitlán,

Cuernavaca, Morelos, México 62100.

Tel: (+52)-777-3293000 ext. 2406

Please find enclosed the cover letter to explain, point by point, to the comments raised by the reviewers.

REVIEWER # 2

Point 1. The reviewer said:

[1] “1. Introduction”, Page 1 of 23, Lines 44-46:

“Cervical cancer remains the fourth most frequently diagnosed malignancy in women and one of the leading causes of cancer-related mortality worldwide, especially in low- and middle-income countries [1].”.

Indeed, despite vaccine policies and screening offered by many countries, cervical cancer remains the fourth leading cause of cancer in women worldwide. At that stage, the authors should report that NHS England first aimed to tackle the burden of the disease by introducing a national cervical screening programme in 1988, which has since seen a significant reduction in over a third of cases in England. Cervical cancer screening is available from the age of 25, as the disease is rare among younger individuals.

Recommended reference: Choi S, et al. HPV and Cervical Cancer: A Review of Epidemiology and Screening Uptake in the UK. Pathogens 2023;12(2):298.

Response to point 1. Thank very much for these comments.

In Introduction section, we included this sentence, and we included the reference [4]:

“Interestingly, National Health Service England first aimed to tackle the burden of the disease by introducing a national cervical screening program in 1988, which has since seen a significant reduction in over a third of cases in England [4]”. Lines 48-56.

[4].      Choi, S.; Ismail, A.; Pappas-Gogos, G.; Boussios, S. HPV and cervical cancer: a review of epidemiology and screening uptake in the UK. Pathogens. 2023, 12(2), 298. DOI: 10.3390/pathogens12020298.

Point 2. The reviewer said:

[2] “1. Introduction”, Page 2 of 23, Lines 49-50:

“Cervical carcinoma has the capacity to metastasize to distant tissues and organs, a fact strongly associated with treatment failure.”.

From the therapeutic perspective, the authors should mention that ongoing immunotherapy trials evaluate vaccine-based therapies, adoptive T-cell therapy, and immune-modulating agents in patients with advanced cervical cancer.

Response to point 2. Thank very much for these comments.

In Introduction section, we included this sentence, and we included the reference [5]:

“However, different research groups ongoing immunotherapy trials to evaluate vaccine-based therapies, adoptive T-cell therapy, and immune-modulating agents in patients with advanced cervical cancer [5]”. Lines 57-59.

  1. US National Institutes of Health [http://www.nlm.nih.gov/copyright.html]. Database of clinical trials. Revision: v2.5.0. US Government, Accessed March 31, 2024. URL: http://clinicaltrials.gov/ct2/home

Point 3. The reviewer said:

[3] “3. Discussion”, Page 11 of 23, Lines 381-385:

“RECK is a serine protease inhibitor that regulates various physiologic and pathologic events. In hepatocellular carcinoma, RECK expression is associated with less vascular invasion, as well as immunogenic features such as recruitment of tumor-infiltrating lymphocytes and expression of immune checkpoint molecules, including programmed cell death ligand 1 (PD-L1) [64].”.

At that point, the authors should report that PD-L1 is more frequently expressed by squamous cell carcinoma than by adenocarcinoma, which accounts for approximately 20–25% of all cervical malignancies.  Diffuse PD-L1 expression in squamous cell carcinoma patients is correlated with poor disease-free survival and disease-specific survival compared with marginal PD-L1 expression, which is associated with a remarkably favorable prognosis. In adenocarcinoma, there is a survival benefit for patients with tumor lacking PD-L1-positive tumor-associated macrophages.

Response to point 3. Thank very much for these comments.

In Discussion section, we included this sentence, and we included the reference [67]:

“RECK is a serine protease inhibitor that regulates various physiologic and pathologic events. In cervical carcinoma, RECK expression is associated with recruitment of tumor-infiltrating lymphocytes and expression of immune checkpoint molecules, including programmed cell death ligand 1 (PD-L1). PD-L1 is more frequently expressed by squamous cell carcinoma than by adenocarcinoma, which accounts for approximately 20–25% of all cervical malignancies.  Diffuse PD-L1 expression in squamous cell carcinoma patients is correlated with poor disease-free survival and disease-specific survival compared with marginal PD-L1 expression, which is associated with a remarkably favorable prognosis. In adenocarcinoma, there is a survival benefit for patients with tumor lacking PD-L1-positive tumor-associated macrophages [67]”. Line 426-435.

[67].    Rotman, J.; den Otter, L.A.S.; Bleeker, M.C.G.; Samuels, S.S.; Heeren, A.M.; Roemer, M.G.M.; Kenter, G.G.; Zijlmans, H.J.M.A.A.; van Trommel, N.E.; de Gruijl, T.D.; Jordanova, E.S. PD-L1 and PD-L2 expression in cervical cancer: regulation and biomarker potential. Front Immunol, 2020, 11, 596825. DOI: 10.3389/fimmu.2020.596825.

Point 4. The reviewer said:

Comments on the Quality of English Language

Minor editing of English language required.

Response to point 4. Thank very much for these comments.

We reviewed the English language editing of manuscript.

All the comments suggested by the Reviewer were included and are indicated in bold in the manuscript.

Reviewer 3 Report

Comments and Suggestions for Authors The manuscript "miR-21 Regulates the RECK Signaling Pathway" by Aguilar-Martínez et al. explores the nexus between the RECK signaling pathway and miR-21. The study presents relevant functional assays demonstrating miR-21's direct control over RECK mRNA and protein expression and its effects on the growth and migration of HPV16+ CaSki cervical carcinoma cells. While the data supports the conclusions drawn, several concerns must be addressed before publication. Major Concerns: 1. The study's primary weakness lies in its reliance on a single HPV+ CaSki cell line. Such a narrow focus raises concerns about potential cell-specific effects observed in the study. It is advisable to replicate the experiments using at least one additional HPV+ SiHa cell line or any other HPV+ cervical cancer cell to validate the results for consistency. Alternatively, the authors should explain why validating the experiments in a second cell line would not impact the study's conclusion. 2. Figure 2b presents densitometric quantification data; however, the method and software used for this quantification are not detailed in the method section. Adding these details would enhance the transparency and reproducibility of the study. 3. the manuscript should depict the miR-21 binding sites within the 3' UTR of RECK mRNA to improve clarity. Additionally, clarifying the size of the 3' UTR is crucial for readers to understand the position of the miR-21 target sites and the overall structure of the UTR. 4. The expression of miR-21-5p in the transfected cell lines from the pSIMIR21-5P overexpression experiments (Figures 2-5) is not addressed. It is essential to assess the mature miRNA expression levels in the transfected cell lines, as differential transfection efficiencies may lead to variable RECK expression levels, as shown in Figure 1A. This data should be provided as supplemental material. 5. Ambiguity arises in lines 336–338 regarding the induction of a migratory phenotype in cervical cancer cells mediated by RECK gene overexpression through miR-21 microRNA silencing. The authors should clarify whether they meant "silencing OF the miR-21 microRNA" or "silencing BY the miR-21 microRNA." 6. Significant issues are observed in the experiment shown in Figure 5B, particularly regarding the differential expression observed in the RECKLuc1, RECKLuc2, and RECKLuc3 experiments and the subsequent results of miR-21-5p suppression of the RECKLuc constructs. The authors should comment on whether this differential expression is influenced by varying lengths of UTRs, sequence context, or secondary local RNA structure. 7. Figure 6A recommends "panel" instead of "lane" for clarity. Furthermore, Figure 6B suffers from lower resolution, making it difficult to interpret the data. Improving the resolution or presenting the data in a more accessible format would enhance its visibility. 8. Figure 6A appears to demonstrate cell proliferation. In this case, conducting the BrDU experiment would be the most appropriate method for demonstrating the effect of miR-21 overexpression on cellular proliferation. 9. Figure 6B shows similar results for 3 µg of U6 and pmiR-21-5p, while only 5 µg shows a measurable effect of pmiR-21-5p. The authors should explain this discrepancy by quantifying the overexpression of mature miR-21 in these experiments.

Author Response

Ms. Tamara Ugarković

Assistant Editor, MDPI Belgrade

MDPI Branch Office, Belgrade

Editorial Office

Int J Mol Sci

Dear Ms. Tamara Ugarković

Thank you very much for giving us the opportunity to publish our manuscript entitled: Manuscript ID: ijms-2887481. Type of manuscript: Article: Title: “Mir-21 regulates growth and migration of cervical cancer cells by RECK signaling pathway”. Seidy Y. Aguilar-Martínez, Gabriela E. Campos-Viguri, Selma E. Medina-García, Ricardo J. García-Flores, Jessica Deas, Claudia Gómez-Cerón, Abraham Pedroza-Torres, Elizabeth Bautista-Rodríguez, Gloria Fernández-Tilapa, Mauricio Rodríguez-Dorantes, Carlos Pérez-Plasencia, Oscar Peralta-Zaragoza *, in Int J Mol Sci., in the original articles section.

We are very interested in re-submitting our manuscript and I am attaching a cover letter to explain, point by point, to the comments raised by the reviewers. In addition, I ensured that all changes in the manuscript are indicated in the text in bold by highlighting or using track changes.

I look forward to receiving your positive response in this regard.

Best regards

Oscar Peralta-Zaragoza. Ph. D.

Direction of Chronic Infections and Cancer,

Research Center in Infection Diseases,

National Institute of Public Health.

Av. Universidad No. 655, Cerrada los Pinos

y Caminera. Colonia Santa María Ahuacatitlán,

Cuernavaca, Morelos, México 62100.

Tel: (+52)-777-3293000 ext. 2406

Please find enclosed the cover letter to explain, point by point, to the comments raised by the reviewers.

REVIEWER # 3

Point 1. The reviewer said:

The manuscript "miR-21 Regulates the RECK Signaling Pathway" by Aguilar-Martínez et al. explores the nexus between the RECK signaling pathway and miR-21. The study presents relevant functional assays demonstrating miR-21's direct control over RECK mRNA and protein expression and its effects on the growth and migration of HPV16+ CaSki cervical carcinoma cells. While the data supports the conclusions drawn, several concerns must be addressed before publication.

Major Concerns

  1. The study's primary weakness lies in its reliance on a single HPV+ CaSki cell line. Such a narrow focus raises concerns about potential cell-specific effects observed in the study. It is advisable to replicate the experiments using at least one additional HPV+ SiHa cell line or any other HPV+ cervical cancer cell to validate the results for consistency. Alternatively, the authors should explain why validating the experiments in a second cell line would not impact the study's conclusion.

Response to point 1. Thank very much for these comments.

We carried out the experiments using one additional HPV16+ cervical cancer cell line (SiHa cells), to validate the consistency of results. Additionally, we modified the manuscript, as well as the figures, to include the results obtained in the SiHa cell line, which were consistent with CaSki cells. Thus, in our study, these data support the potential effects observed in RECK downregulation mediated by silencing of miR-21 microRNA in HPV16+ cervical cancer cells.

Point 2. The reviewer said:

  1. Figure 2b presents densitometric quantification data; however, the method and software used for this quantification are not detailed in the method section. Adding these details would enhance the transparency and reproducibility of the study.

Response to point 2. Thank very much for these comments.

In Material al Methos section, in “semiquantitative end-point RT-PCR analysis of RECK gene” section, we included the next sentence:

“PCR product bands were digitalized and analyzed by densitometer using ImageJ which is a Java-based image processing program. ImageJ program can display, edit, analyze, process, save, and print 8-bit color and grayscale, 16-bit integer, and 32-bit floating point images. It can read many image file formats, including TIFF, PNG, GIF, JPEG, BMP, DICOM, and FITS, as well as raw formats.” Line 696-701.

Additional, in the legend of figures 2 and 4, we change the sentence:

“bands were digitalized and analyzed by densitometer…”

By the sentence:

bands were digitalized and analyzed by densitometer using ImageJ program”. Line 189-190, and Line 248-249.

Point 3. The reviewer said:

  1. The manuscript should depict the miR-21 binding sites within the 3' UTR of RECK mRNA to improve clarity. Additionally, clarifying the size of the 3' UTR is crucial for readers to understand the position of the miR-21 target sites and the overall structure of the UTR.

Response to point 3. Thank very much for these comments.

In the first version of manuscript, in Material and Methods section, we depict the miR-21 binding sites within the 3' UTR of RECK gene.

“CaSki and SiHa cells were transiently transfected with pMRE21RECKLuc1 plasmid which contain cloned the MRE21-1 (microRNA response element 1 for miR-21 located of 3343 to 3366 nt), with pMRE21RECKLuc2 plasmid which contain cloned the MRE21-2 (microRNA response element 2 for miR-21 located of 3893 to 3918 nt), and with pMRE21RECKLuc3 plasmid which contain cloned the MRE21-3 (microRNA response element 3 for miR-21 located of 4039 to 4061 nt) of RECK 3′-UTR regulatory region.” Line 754-759.

In Material and Methods section, we included the next sentence:

“The RECK 3’-UTR regulatory region sequence downstream of the transcriptional start site of RECK gene has a longitude of 1410 nt (from 2917 nt to 4326 nt).” Line 761-763.

Point 4. The reviewer said:

  1. The expression of miR-21-5p in the transfected cell lines from the pSIMIR21-5P overexpression experiments (Figures 2-5) is not addressed. It is essential to assess the mature miRNA expression levels in the transfected cell lines, as differential transfection efficiencies may lead to variable RECK expression levels, as shown in Figure 1A. This data should be provided as supplemental material.

Response to point 4. Thank very much for these comments.

Previously, the expression of miR-21 in the transfected cell lines with pSIMIR21-5P was reported by our group research [45].

However, to address this point, in Material and Methods section, we included the next sentence:

“Previously, we generated and demonstrated that pSIMIR21-5P is a siRNA expression plasmid specific for miR-21, which has the ability to induce selective and specific silencing of miR-21 microRNA in human cervical cancer cells infected with HPV16 [50, 81]. Line 652-655.

[50]. Peralta-Zaragoza, O.; Deas, J.; Meneses-Acosta, A.; De la O-Gómez, F.; Fernández-Tilapa, G.; Gómez-Cerón, C.; Benítez-Boijseauneau, O.; Burguete-García, A.; Torres-Poveda, K.; Bermúdez-Morales, V.H.; Madrid-Marina, V.; Rodríguez-Dorantes, M.; Hidalgo-Miranda, A, Pérez-Plasencia, C. Relevance of miR-21 in regulation of tumor suppressor gene PTEN in human cervical cancer cells. BMC Cancer. 2016, 16, 215. DOI: 10.1186/s12885-016-2231-3.

[81].    Peralta-Zaragoza, O.; De-la-O-Gómez, F.; Deas, J.; Fernández-Tilapa, G.; Fierros-Zárate, G.del S.; Gómez-Cerón, C.; Burguete-García, A.; Torres-Poveda, K.; Bermúdez-Morales, V.H.; Rodríguez-Dorantes, M.; Pérez-Plasencia, C.; Ma-drid-Marina, V. Selective silencing of gene target expression by siRNA expression plasmids in human cervical cancer cells. Methods Mol Biol. 2015, 1249, 153-71. DOI: 10.1007/978-1-4939-2013-6_11.

Point 5. The reviewer said:

  1. Ambiguity arises in lines 336–338 regarding the induction of a migratory phenotype in cervical cancer cells mediated by RECK gene overexpression through miR-21 microRNA silencing. The authors should clarify whether they meant "silencing OF the miR-21 microRNA" or "silencing BY the miR-21 microRNA."

Response to point 5. Thank very much for these comments.

In Results, in “Silencing of mir-21 microRNA by siRNAs induces alterations in tumor cell migration” section, we change the sentence:

“This date suggests a differential migratory phenotype in cervical cancer cells that may be induced by RECK gene overexpression mediated by silencing miR-21 microRNA.”

By the sentence:

“This date suggests a differential migratory phenotype in cervical cancer cells that may be induced by RECK gene overexpression mediated by silencing of miR-21 microRNA.” Line 372-374.

Point 6. The reviewer said:

  1. Significant issues are observed in the experiment shown in Figure 5B, particularly regarding the differential expression observed in the RECKLuc1, RECKLuc2, and RECKLuc3 experiments and the subsequent results of miR-21-5p suppression of the RECKLuc constructs. The authors should comment on whether this differential expression is influenced by varying lengths of UTRs, sequence context, or secondary local RNA structure.

Response to point 6. Thank very much for these comments.

In Results, in “Specific MRE recognition sequences by mir-21 are critical for regulation of RECK” section, we included the next sentence:

“Furthermore, when CaSki and SiHa cells were transfected with pMRE21RECKLuc1, pMRE21RECKLuc2, and pMRE21RECKLuc3, and co-transfected with pSIMIR21-5P plasmids, we observed a differential expression of the luciferase activity. This differential expression could be influenced by varying lengths of RECK 3’-UTRs, by sequence context of nucleotide complementary between the seed sequence of MRE21-1, MRE21-2, or MRE21-3 with RECK 3’-UTR complementary sequence, or by secondary local RNA structure.” Line 287-293.

Point 7. The reviewer said:

  1. Figure 6A recommends "panel" instead of "lane" for clarity. Furthermore, Figure 6B suffers from lower resolution, making it difficult to interpret the data. Improving the resolution or presenting the data in a more accessible format would enhance its visibility.

Response to point 7. Thank very much for these comments.

In Results, in “Silencing of mir-21 microRNA by siRNAs induces alterations in tumor cell migration” section, we changed the sentence:

“Figures 6A and 6B, show the effect on cellular proliferation and viability after no transfection (Figure 6A, NT, lanes a, b, c, d, e, and f), transfection with pSilencer 1.0-U6 empty vector plasmid (Figure 6A, lanes aa-cf) and transfection with pSIMIR21-5P plasmid (Figure 6A, lanes da-ff).”

By the sentence:

“Figures 6A and 6B, show the effect on cellular proliferation and viability after no transfection (Figure 6A, NT, panel a, b, c, d, e, and f), transfection with pSilencer 1.0-U6 empty vector plasmid (Figure 6A, panel aa-cf) and transfection with pSIMIR21-5P plasmid (Figure 6A, panel da-ff).” Line 329-331.

We changed the sentence:

“The results show that after 24 h of transfection with pSIMIR21-5P plasmid, silencing miR-21 induced decreased migration of CaSki cells (Figure 7A, lanes ah and al) compared with non-transfected CaSki cells (Figure 7A, lanes a-d and aa-ad) or transfected with pSilencer 1.0-U6 empty vector plasmid (Figure 7A, lanes e-l).”

By the sentence:

“The results show that after 24 h of transfection with pSIMIR21-5P plasmid, silencing miR-21 induced decreased migration of CaSki cells (Figure 7A, panel ah and al) compared with non-transfected CaSki cells (Figure 7A, panel a-d and aa-ad) or transfected with pSilencer 1.0-U6 empty vector plasmid (Figure 7A, panel e-l).” Line 352-355.

We changed the sentence:

“The results show that after 24 h of transfection with pSIMIR21-5P plasmid, silencing miR-21 induced decreased migration of SiHa cells (Figure 8A, lanes e-f) compared with transfected SiHa cells with pSilencer 1.0-U6 empty vector plasmid (Figure 8A, lanes a-d).”

By the sentence:

“The results show that after 24 h of transfection with pSIMIR21-5P plasmid, silencing miR-21 induced decreased migration of SiHa cells (Figure 8A, panel e-f) compared with transfected SiHa cells with pSilencer 1.0-U6 empty vector plasmid (Figure 8A, panel a-d).” Line 362-365.

Additionally, the data of Figure 6B, were presenting in a more accessible format to enhance its visibility. We changed the data to color.

Point 8. The reviewer said:

  1. Figure 6A appears to demonstrate cell proliferation. In this case, conducting the BrDU experiment would be the most appropriate method for demonstrating the effect of miR-21 overexpression on cellular proliferation.

Response to point 8. Thank very much for these comments.

The cell proliferation can be measured with the thymidine analog BrDU (5-bromo-2’-deoxyuridine) following its incorporation into newly synthesized DNA and its subsequent detection with an anti-BrDU antibody.

However, in our study, in Material and Methods section, we described the cellular viability assays using the MTS method.

“Cellular viability was measured using [3-(4,5-dimethylthiazol-2-yl)-5-(3-carboxymethoxyphenyl)-2-(4-sulfophenyl)-2H-tetrazolium] inner salt MTS assay (Promega, Madison WI), which is a colorimetric method for determining the number of viable cells in a proliferation or cytotoxicity assay…….” Line 799-800.

Point 9. The reviewer said:

  1. Figure 6B shows similar results for 3 µg of U6 and pmiR-21-5p, while only 5 µg shows a measurable effect of pmiR-21-5p. The authors should explain this discrepancy by quantifying the overexpression of mature miR-21 in these experiments.

Response to point 9. Thank very much for these comments.

In Discussion section, we included the next sentence:

“The Figure 6B shows similar results of proliferation cells when CasKi cells were transfected with 3 µg of pSilencer 1.0-U6 and PSIMIR21-5P, while transfection with 5 µg of PSIMIR21-5P we observed a measurable effect of proliferation cell. This discrepancy in the proliferation cell could be explained in part by endogenous miR-21 overexpression which is approximately 3 time more in CaSki cells that in HaCaT cells (Figure 1A).” Line 613-618.

All the comments suggested by the Reviewer were included and are indicated in bold in the manuscript.

Reviewer 4 Report

Comments and Suggestions for Authors

I appreciate the opportunity to make second review  of the manuscript entitled “Mir-21 regulates growth and migration of cervical cancer cells by RECK signaling pathway.” submitted in the journal International Journal of Molecular Medicine.

Reviewers comments:

1.      The manuscript include numerous abbereviations ,please make the list of the abbreviations and included it in the manuscript.

2.      Please add the paragraphs of the potential diagnostic/therapeutic usage of the results of the yours manuscript.

My opinion is that this submission meets the criteria to be published in the journal International Journal of Molecular Medicine after this revision I suggested.

Author Response

Ms. Tamara Ugarković

Assistant Editor, MDPI Belgrade

MDPI Branch Office, Belgrade

Editorial Office

Int J Mol Sci

Dear Ms. Tamara Ugarković

Thank you very much for giving us the opportunity to publish our manuscript entitled: Manuscript ID: ijms-2887481. Type of manuscript: Article: Title: “Mir-21 regulates growth and migration of cervical cancer cells by RECK signaling pathway”. Seidy Y. Aguilar-Martínez, Gabriela E. Campos-Viguri, Selma E. Medina-García, Ricardo J. García-Flores, Jessica Deas, Claudia Gómez-Cerón, Abraham Pedroza-Torres, Elizabeth Bautista-Rodríguez, Gloria Fernández-Tilapa, Mauricio Rodríguez-Dorantes, Carlos Pérez-Plasencia, Oscar Peralta-Zaragoza *, in Int J Mol Sci., in the original articles section.

We are very interested in re-submitting our manuscript and I am attaching a cover letter to explain, point by point, to the comments raised by the reviewers. In addition, I ensured that all changes in the manuscript are indicated in the text in bold by highlighting or using track changes.

I look forward to receiving your positive response in this regard.

Best regards

Oscar Peralta-Zaragoza. Ph. D.

Direction of Chronic Infections and Cancer,

Research Center in Infection Diseases,

National Institute of Public Health.

Av. Universidad No. 655, Cerrada los Pinos

y Caminera. Colonia Santa María Ahuacatitlán,

Cuernavaca, Morelos, México 62100.

Tel: (+52)-777-3293000 ext. 2406

Please find enclosed the cover letter to explain, point by point, to the comments raised by the reviewers.

REVIEWER # 4

Point 1. The reviewer said:

I appreciate the opportunity to make second review  of the manuscript entitled “Mir-21 regulates growth and migration of cervical cancer cells by RECK signaling pathway.” submitted in the journal International Journal of Molecular Medicine.

Comments and Suggestions for Authors

Reviewer’s comments:

  1. The manuscript include numerous abbreviations, please make the list of the abbreviations, and included it in the manuscript.

Response to point 1. Thank very much for these comments.

We included in manuscript the next list of abbreviations:

Abbreviations. Line 888-925.

ACAT1: Acetyl-CoA acetyltransferase 1.

ATCC: American Type Culture Collection.

BCA kit: (Bicinchoninic Acid) Protein Assay kit.

BTG2: Protein BTG2 also known as BTG family member 2 or NGF-inducible anti-

proliferative protein PC3 or NGF-inducible protein TIS21.     

CDH1: Cadherin-1 or Epithelial cadherin (E-cadherin).

CDK6: Ccyclin-Dependent Kinase 6.

CRISPR: Clustered Regularly Interspaced Short Palindromic Repeats.

DEPC: Diethylpyrocarbonate.

DMEM: Dulbecco’s modified Eagle’s medium.

EDTA: Ethylenediaminetetraacetic Acid.

FBS: Fetal Bovine Serum.

GAPDH: Glyceraldehyde 3-phosphate dehydrogenase.

GAS5: Growth arrest-specific 5.

GPI: Glycosylphosphatidylinositol.

HIF1A: Hypoxia-inducible factor 1 alpha.

HPV: Human papilloma virus.

IDT: Integrated DNA Technologies.

ITGB1: Integrin beta-1, also known as CD29.

ITGB3: Integrin beta-3, also known as CD61.

kDa: Kilo Dalton.

MARCKS: Myristoylated alanine-rich C-kinase substrate.

MMPs: Matrix metalloproteinases.

MRE: MicroRNA Response Element.

MTS: [3-(4,5-dimethylthiazol-2-yl)-5-(3-carboxymethoxyphenyl)-2-(4-sulfophenyl)-2H-tetrazolium].

PDCD4: Programmed cell death protein 4.

PEI: Polyethylenimine linear MW 25,000 transfection grade

PMSF: Phenylmethylsulfonyl fluoride.

PTEN: Phosphatase and tensin homolog.

RECK: Reversion-inducing-cysteine-rich protein with kazal motifs.

RT-PCR: Reverse transcription polymerase chain reaction.

RT-qPCR : Reverse transcription quantitative polymerase chain reaction.

SCC: Squamous cell carcinoma.

siRNAs: Small interfering RNA, also know as short interfering RNA, or silencing RNA.

TPM1: Tropomyosin 1.

Point 2. The reviewer said:

  1. Please add the paragraphs of the potential diagnostic/therapeutic usage of the results of the yours manuscript.

Response to point 1. Thank very much for these comments.

In Conclusion section, we included the next sentence:

“Our findings suggest that a therapeutic strategy employing siRNAs could effectively inhibit the growth of virus-related cancers. Furthermore, RECK could be a promising prognostic biomarker and may shape a low-metastasis microenvironment while miR-21 may shape high-tumor-progression in patients with cervical cancer.” Line 855-858.

Point 3. The reviewer said:

My opinion is that this submission meets the criteria to be published in the journal International Journal of Molecular Medicine after this revision I suggested.

Response to point 1. Thank very much for these comments.

All the comments suggested by the Reviewer were included and are indicated in bold in the manuscript.
